



# Error-correction across gauged and ungauged locations: A data assimilation-inspired approach to post-processing river discharge forecasts

Gwyneth Matthews[1,2], Hannah L Cloke[1,3], Sarah L Dance[1,4,5], and Christel Prudhomme[2]

[1]Department of Meteorology, University of Reading, Reading, United Kingdom
[2]European Centre for Medium-range Weather Forecasts, Reading, United Kingdom
[3]Department of Geography and Environmental Science, University of Reading, Reading, United Kingdom
[4]Department of Mathematics and Statistics, University of Reading, Reading, United Kingdom
[5]National Centre for Earth Observation (NCEO), Reading, United Kingdom

**Correspondence:** Gwyneth Matthews (g.r.matthews@pgr.reading.ac.uk)

**Abstract.** Forecasting river discharge is essential for disaster risk reduction and water resource management, but forecasts of future river state often contain errors. Post-processing reduces forecast errors but is usually only applied at the locations of river gauges, leaving the majority of the river network uncorrected. Here, we present a data-assimilation-inspired method for error-correcting ensemble simulations across gauged and ungauged locations in a post-processing step. Our new method

employs state augmentation within the framework of the Localised Ensemble Transform Kalman Filter (LETKF) to estimate an error vector for each ensemble member. The LETKF uses ensemble error covariances to spread observational information from gauged to ungauged locations in a dynamic and computationally effcent manner. To improve the efficiency of the LETKF we define new localisation, covariance inflation, and initial ensemble generation techniques that can be easily transferred between modelling systems and river catchments. We implement and evaluate our new error-correction method for the entire Rhine-

Meuse catchment using forecasts from the Copernicus Emergency Management Service's European Flood Awareness System (EFAS). The resulting river discharge ensembles are error-corrected at every grid box but remain spatially and temporally consistent. The skill is evaluated at 89 proxy-ungauged locations to assess the ability of the method to spread the correction along the river network. The skill of the ensemble mean is improved at almost all locations including stations both up- and downstream of the assimilated observations. Whilst the ensemble spread is improved at short lead-times, at longer lead-times

the ensemble spread is too large leading to an underconfident ensemble. In summary, our method successfully propagates error information along the river network, enabling error correction at ungauged locations. This technique can be used for improved post-event analysis and can be developed further to post-process operational forecasts providing more accurate knowledge about the future states of rivers.



## 1 Introduction

River discharge forecasts are essential tools for taking effective preparatory actions for disaster mitigation and water resource planning (World Meteorological Organisation (WMO), 2015). However, despite the increased sophistication of forecasting systems over the past few decades, river discharge forecasts still contain uncertainty (Boelee et al., 2019). The uncertainty is introduced at several stages of the forecasting system including the meteorological forcings, the initial conditions, and

the hydrological model structure and parameters (Valdez et al., 2022). Ensemble river discharge forecasts aim to account for the meteorological uncertainty by forcing a hydrological model with many meteorological forcings either from multiple numerical weather prediction (NWP) systems or from an ensemble weather forecast created using multiple sets of initial conditions (Cloke and Pappenberger, 2009; Wu et al., 2020). However, ensemble forecasts can still contain errors. Different methods for correcting these errors have been developed including pre-processing of the meteorological forcings, calibration

of the hydrological model, improving the initial conditions using data assimilation, and post-processing of the river discharge forecast (Bourdin et al., 2012). Of these approaches post-processing is often considered the most computationally efficient and its ability to correct for multiple sources of errors simultaneously is appealing.

In meteorological forecasting, post-processing at non-observed locations is common (see Vannitsem et al., 2021). However, hydrological forecasting also requires consideration of the spatial heterogeneity introduced by the river network (e.g., Li et al.,

2017; Woldemeskel et al., 2018; Ye et al., 2014; Xu et al., 2019; Liu et al., 2022; Lee and Ahn, 2024) and the application of post-processing methods at ungauged locations is still a difficult challenge. The lack of gauged locations along river networks is a particular problem as is the lack of agreed data sharing practices for the areas that are gauged (Lavers et al., 2019; Hannah et al., 2011), which means that the development of post-processing techniques for ungauged locations is essential. However, current techniques are generally too computationally expensive for operational river flow forecasting applications

(Emerton et al., 2016). For example, defining a joint distribution between the river discharge at multiple locations would allow forecasts to be conditioned on observations available at specific locations (Engeland and Steinsland, 2014). However, for large-scale distributed systems and multiple lead-times the size of the joint distribution quickly becomes too large. Alternatively, error-correction can be performed at gauged location and the results interpolated to ungauged locations. One such method used to interpolate error-correction parameters is top-kriging (Pugliese et al., 2018; Skøien et al., 2021). Top-kriging takes

into account the river network but the relationship between errors at different locations is assumed static regardless of the hydrometeorological situation (Skøien et al., 2016, 2006). Another option is to use a river routing model to propagate error-corrected river discharge forecasts between gauged locations using a river routing model (Bennett et al., 2022). Whilst this approach maintains spatial consistency between locations, the additional run of the model could be computationally expensive for an operational application.

The aim of this paper is to present and evaluate a novel technique to spread observation information from gauged to un-gauged locations in a computationally efficient and temporally varying manner. The new method is based on data assimilation techniques, commonly used to improve the initial conditions of forecasts (Valdez et al., 2022), but applied as post-processing so that additional, computationally expensive executions of the hydrological model are not required. Data assimilation is a





mathematical technique that combines modelled predictions and observations to produce an improved modelled state relative

to the true state of the system (Nichols, 2003, 2010). The error correction method proposed in this study is based on the Local
Ensemble Transform Kalman Filter (LETKF, Hunt et al., 2007) and state augmentation (Dee, 2005). The LETKF is part of
the Kalman Filter family of methods and uses an ensemble of model states to estimate the state error covariances. Due to its
computational efficiency and ability to handle non-linear dynamics without an adjoint model, ensemble Kalman Filters are
common data assimilation methods for hydrological applications (Rouzies et al., 2024; Li et al., 2023; Mason et al., 2020;

Ridler et al., 2018; Khaki et al., 2017; Xie and Zhang, 2010; Clark et al., 2008). State augmentation is a technique that allows
the estimation of the state of a system and the parameters of the model used to simulate that system simultaneously (Ridler
et al., 2018; Gharamti and Hoteit, 2014; Smith et al., 2013, 2009; Martin et al., 2002).

The proposed method aims to improve the skill of the ensemble mean and the reliability of the ensemble spread by adjusting
each ensemble member, as will be discussed in more detail in Section 2. However, it is equally if not more important that the

ensembles are spatially and temporally consistent in order to aid with decision making (Bennett et al., 2022). This is particularly
important for large scale systems that provide forecasts across administrative boundaries, such as the Copernicus Emergency
Management Service's (CEMS) European Flood Awareness System (EFAS) used in this study (Matthews et al., 2025). The
specific research questions to be addressed in this study are therefore,

1. Can data assimilation techniques be used in a post-processing environment to spread observation information to un-
gauged locations in a spatiotemporally consistent manner?

2. Are the resulting ensemble predictions of river discharge more skillful than the raw ensemble?

This paper is organised as follows. In Section 2 we define the errors which we aim to correct and introduce some terminology
and notation. In Section 3 we formulate the data assimilation techniques used within this study. In Section 4 we outline the
proposed error-correction method and detail how the ensemble is corrected. Section 5 provides some additional components of

the method that improve the efficacy of the method but can be adjusted to suit the data availability of any system and/or domain.
Section 6 outlines the strategy used to evaluate the efficacy of the proposed method and Section 7 presents the results, first
assessing the ability of the method to spread observational information to ungauged locations and then assessing the skill of the
error-corrected ensembles. In Section 8 we discuss key features of the proposed method and their impact on the error-corrected
ensembles. In Section 9 we conclude that the proposed method successfully improves the skill of the ensemble and maintains

spatiotemporal consistency, and highlight priorities for future developments.

Please note that throughout the paper 'hindcast ensemble' refers to the ensembles of river discharge that we are error-
correcting. These ensembles are past operational EFAS forecasts (see Section 6.1); however, when we perform the error-
correction we use observations that are available within the forecast (hindcast) period which would not be possible in an
operational system as these timesteps would be in the future. Therefore, we refer to these ensembles as hindcasts for clarity.



## 2 Ensemble error-correction framework

Here, we define the errors which we aim to correct and provide some notation that is used throughout the paper. Let the true state of the system at time $k$ be defined as $\mathbf{x}_k^{true} \in \mathbb{R}^n$, where each element represents the true river discharge in one of the $n$ grid boxes in the domain of interest. Hydrological forecasts, including the EFAS forecasts used in this study (Section 6.1), generally estimate the true state of the system by a modelled state, denoted $\mathbf{x}_k$ where the lack of superscript 'true' indicates it is a modelled estimate. Hydrological ensemble forecasts consist of $N$ potential realizations referred to as ensemble members. We define the ensemble river discharge hindcasts used in this study as

$$\left\{ \mathbf{x}_k : \mathbf{x}_k^{(i)}, \text{for } i = 1, 2, \ldots, N \text{ and } k = 0, 1, \ldots, L \right\}. \tag{1}$$

where the superscript $(i)$ indicates the $i$-th member of the ensemble, $N$ is the ensemble size, the timestep $k$ refers to the lead-time of the hindcast, and $L$ is the maximum lead-time. The ensemble mean is defined as

$$\overline{\mathbf{x}}_k = \frac{1}{N} \sum_{i=1}^{N} \mathbf{x}_k^{(i)} \in \mathbb{R}^n. \tag{2}$$

The ensemble perturbation matrix is defined as

$$\mathbf{X}_k = \left( \begin{array}{cccc} \mathbf{x}_k^{(1)} - \overline{\mathbf{x}}_k & \mathbf{x}_k^{(2)} - \overline{\mathbf{x}}_k & \cdots & \mathbf{x}_k^{(N)} - \overline{\mathbf{x}}_k \end{array} \right) \in \mathbb{R}^{n \times N} \tag{3}$$

where the $i$-th column represents the $i$-th ensemble member's departure from the ensemble mean at lead-time $k$. The perturbation matrix contains information about the spread of the ensemble and the spatial structure of the deviations from the mean of each ensemble member. From the definition of the perturbation matrix, the ensemble covariance matrix is defined as

$$\mathbf{P}_k = \frac{1}{N-1} \mathbf{X}_k \mathbf{X}_k^T \in \mathbb{R}^{n \times n}. \tag{4}$$

where the superscript $T$ indicates the matrix transpose.

Hydrological ensembles may still contain errors, so a post-processing step is usually necessary within a hydrological forecasting system. In this paper, we propose a method to spread an error-correction from gauged locations to every grid box in the system domain. We assume that there is an additive relationship between each hindcast ensemble member and an error vector such that

$$\mathbf{x}_k^{true} = \mathbf{x}_k^{(i)} + \mathbf{b}_k^{(i)true} \in \mathbb{R}^n \tag{5}$$

where $\mathbf{b}_k^{(i)true}$ is the error of the $i$-th hindcast ensemble member with respect to the true state. Each element of the error vector is the error associated with a single grid box. The proposed method estimates the additive error vector, $\mathbf{b}_k^{(i)}$ (where the lack of the superscript *true* indicates it is an estimate) for each hindcast ensemble member at each timestep $k$ such that the resulting hindcast distribution, $\mathbf{x}_k^{new}$, is defined by

$$\mathbf{x}_k^{new} \sim \mathcal{N}(\overline{\mathbf{x}}_k + \overline{\mathbf{b}}_k, \mathbf{P}_k + \mathbf{\Gamma}_k) \tag{6}$$





where $\overline{\mathbf{x}}_k \in \mathbb{R}^n$ and $\mathbf{P}_k \in \mathbb{R}^{n \times n}$ are the ensemble mean and the ensemble covariance matrix of the raw ensemble, respectively, $\overline{\mathbf{b}}_k \in \mathbb{R}^n$ is the ensemble mean of the estimated error vectors, and $\mathbf{\Gamma}_k$ is an additive spread correction matrix. Defining an error-corrected ensemble in terms of mean bias and spread correction parameters is a common post-processing technique used, for example, in the Ensemble Model Output Statistics (EMOS Gneiting et al., 2005; Skøien et al., 2021) method.

To aid with the estimation of the error vectors, we assume that at each timestep the system is observed at $p_k$ river discharge gauges such that we have a vector of river discharge observations, $\mathbf{y}_k \in \mathbb{R}^{p_k}$. We assume the observation vector is related to the true state of the system as

$$\mathbf{y}_k = \mathbf{H}_k(\mathbf{x}_k^{true}) + \boldsymbol{\epsilon}_k \tag{7}$$

where $\boldsymbol{\epsilon}_k \in \mathbb{R}^{p_k}$ is a vector of unbiased Gaussian noise with covariance matrix $\mathbf{R}_k \in \mathbb{R}^{p_k \times p_k}$, such that $\boldsymbol{\epsilon}_k \sim \mathcal{N}(\mathbf{0}, \mathbf{R}_k)$, and $\mathbf{H}_k \in \mathbb{R}^{p_k \times n}$ is the linear observation operator which maps the variables from the state space to observation space. The observation operator used in this study selects the grid boxes from the modelled drainage network of the hydrological model that represent the location of the river gauges.

## 3 Data Assimilation

As discussed in Section 1, the proposed method is based on common data assimilation techniques: state augmentation and the Local Ensemble Transform Kalman Filter (LETKF). In this section we present the formulations of these techniques used in this study.

### 3.1 State augmentation

In the proposed method, an ensemble of augmented states is defined between the ensemble river discharge hindcast (see Section 2) and an ensemble of additive error vectors. We define this ensemble of error vectors at time $k$ as

$$\left\{ \mathbf{b}_k^{(i)} \in \mathbb{R}^n \text{ for } i = 1, 2, \ldots, N \right\} \tag{8}$$

where $N$ is the same ensemble size as the river discharge hindcast and $n$ is the number of grid-boxes in the hindcast domain. The error ensemble mean, $\overline{\mathbf{b}}$, and the ensemble perturbation matrix, $\mathbf{B}$, are calculated by substituting $\mathbf{b}_k^{(i)}$ in place of $\mathbf{x}_k^{(i)}$ in Eqs. (2) and (3), respectively. The generation of the initial error ensemble for timestep $k = 1$ is described in Section 5.3.

The ensemble of augmented states is then defined such that the $i$-th ensemble member is defined as

$$\mathbf{w}_k^{(i)} = \begin{pmatrix} \mathbf{x}_k^{(i)} \\ \mathbf{b}_k^{(i)} \end{pmatrix} \in \mathbb{R}^{2n}. \tag{9}$$

where $\mathbf{x}^{(i)} \in \mathbb{R}^n$ and $\mathbf{b}^{(i)} \in \mathbb{R}^n$ are the $i$-th hindcast and error ensemble members, respectively. The augmented ensemble mean and perturbation matrix are given by

$$\overline{\mathbf{w}}_k = \begin{pmatrix} \overline{\mathbf{x}}_k \\ \overline{\mathbf{b}}_k \end{pmatrix} \in \mathbb{R}^{2n} \quad \text{and} \quad \mathbf{W}_k = \begin{pmatrix} \mathbf{X}_k \\ \mathbf{B}_k \end{pmatrix} \in \mathbb{R}^{2n \times N} \tag{10}$$





where $\overline{\mathbf{x}}$ and $\overline{\mathbf{b}}$ are the ensemble means of the hindcast and error ensembles, respectively, and $\mathbf{X}$ and $\mathbf{B}$ are the perturbation matrices of the hindcast and error ensembles, respectively.

The next step of state augmentation is to define the evolution of the augmented states between timesteps. The hindcast is evolved by the LISFLOOD hydrological model (the hydrological model used to create the EFAS forecasts; Van Der Knijff et al., 2010). For the evolution of the error vectors, we adopt the common assumption that the error is constant between timesteps (Martin, 2001), such that

$$\mathbf{b}_k^{(i)} = \mathbf{b}_{k-1}^{(i)}. \tag{11}$$

Based on these independent evolution equations and the additive relationship between the hindcast ensemble members and the error ensemble members (see Eq. (5)), we define the propagation of the augmented ensemble members as

$$\mathbf{w}_k^{(i)} = \begin{pmatrix} \mathbf{M}_{k-1} & \mathbf{I}_{k-1}^{n \times n} \\ \mathbf{0}_{k-1} & \mathbf{I}_{k-1}^{n \times n} \end{pmatrix} \begin{pmatrix} \mathbf{x}_{k-1}^{(i)} \\ \mathbf{b}_{k-1}^{(i)} \end{pmatrix} = \begin{pmatrix} \mathbf{x}_k^{(i)} + \mathbf{b}_{k-1}^{(i)} \\ \mathbf{b}_{k-1}^{(i)} \end{pmatrix}. \tag{12}$$

where $\mathbf{M}_{k-1} \in \mathbb{R}^{n \times n}$ is a linear evolution operator representing the LISFLOOD hydrological model, $\mathbf{I}_{k-1}^{n \times n}$ is the identity matrix acting on the error component of the augmented state, and $\mathbf{x}_k^{(i)} \in \mathbb{R}^n$ is the $i$-th member of the precomputed the hindcast ensemble. Since we use precomputed hindcast ensembles the propagation of the hindcast ensemble members requires no additional computation and the full non-linear LISFLOOD hydrological model is used without the need to define a linear approximation.

## 3.2 Local Ensemble Transform Kalman Filter (LETKF)

The Local Ensemble Transform Kalman Filter (LETKF, Hunt et al. (2007)) updates the mean state and the square root of the covariance matrix of an ensemble (i.e., the perturbation matrix) by combining the modelled and observed data. As a sequential data assimilation method, the LETKF consists of a *propagation step* (also known as a *forecast step*) and an *update step* (also known as an *analysis step*) that are iterated. We use the LETKF to update the ensemble of error vectors at each timestep but we modify the propagation step to use precomputed hindcasts. The propagation step evolves the augmented states forward in time from time $k-1$ to $k$, as described in Eq. (12). Rather than evolve the hindcast ensemble explicitly (which would require the hydrological model) we instead substitute the precomputed hindcast ensemble for timestep $k$ into the propagated augmented state. The update step of the LETKF calculates the optimal estimate of the state of the system at timestep $k$ by combining the modelled augmented states and observations, both weighted by their respective uncertainties represented by their covariance matrices. As the LETKF is a well documented method we only provide the key update equations and direct the reader to Hunt et al. (2007) and Livings et al. (2008) for more detailed derivations.

To apply the LETKF to the augmented ensemble we extend the definition of the observation operator, $\mathbf{H} \in \mathbb{R}^{p \times n}$, given in Eq. (7) such that

$$\widehat{\mathbf{H}}\mathbf{w}_k^{(i)} = (\mathbf{H}_k \ \ \mathbf{0}) \begin{pmatrix} \mathbf{x}_k^{(i)} + \mathbf{b}_{k-1}^{(i)} \\ \mathbf{b}_{k-1}^{(i)} \end{pmatrix} = \mathbf{H}_k \mathbf{x}_k^{(i)} + \mathbf{H}_k \mathbf{b}_{k-1}^{(i)} \tag{13}$$



where $\widehat{\mathbf{H}} \in \mathbb{R}^{p \times 2n}$ is the augmented observation operator. As discussed in Section 2, the observation operator maps state variables from state space to observation space by extracting data for the appropriate grid-boxes. The LETKF can then update the augmented ensemble mean, $\overline{\mathbf{w}}_k$, such that,

$$\overline{\mathbf{w}}_k^a = \overline{\mathbf{w}}_k^f + \begin{pmatrix} \mathbf{K}_{\mathbf{x}_k} \\ \mathbf{K}_{\mathbf{b}_k} \end{pmatrix} (\mathbf{y}_k - \widehat{\mathbf{H}}_k \overline{\mathbf{w}}_k^f), \tag{14}$$

where the subscripts $f$ and $a$ indicate the state before and after the update step, respectively; $\mathbf{K}_{\mathbf{x}_k} \in \mathbb{R}^{n \times p}$ and $\mathbf{K}_{\mathbf{b}_k} \in \mathbb{R}^{n \times p}$ are the components of the Kalman gain matrix acting on the hindcast ensemble and the error ensemble respectively; and $\mathbf{y}_k \in \mathbb{R}^p$ is the observation vector defined in Eq. (7). The difference between the observations and the model state in observation space (i.e., $\mathbf{y}_k - \widehat{\mathbf{H}}_k \overline{\mathbf{w}}_k^f$) is called the innovation vector. The Kalman gain matrix weights the prior modelled state and the observations based on their respective uncertainties and determines the impact of the innovation vector in the update step. Large observation

uncertainties reduce the Kalman gain, while large uncertainties in the prior state increase the Kalman gain. Both the hindcast and the error components of the Kalman gain are functions of the covariance matrix of the augmented ensemble (see Eqs. (8) and (9) in Bell et al., 2004). The covariance matrix describes the state error covariances between grid-boxes allowing the Kalman gain to spread the observation information to ungauged locations. To update the error component specifically, it is the cross-covariances between the error component and the hindcast component that control the spread of the observation

information to ungauged locations (see Eq. (9) in Bell et al., 2004). This ability to spread the observational information is key to the error-correction method presented in this study.

The LETKF updates the augmented ensemble perturbation matrix, $\mathbf{W}_k$, such that,

$$\mathbf{W}_k^a = \mathbf{W}_k^f \mathbf{T}_k \tag{15}$$

where $\mathbf{T}_k \in \mathbb{R}^{N \times N}$ is the square root transform matrix (Livings et al., 2008). The square root transform matrix is derived

using the Kalman gain matrix which gives the weighting between the modelled state and the observations (Livings et al., 2008). Using an eigenvector decomposition, the square root transform matrix rescales and rotates the ensemble members such that the updated perturbation matrix represents the uncertainty in the updated ensemble mean. The square root transform matrix allows the covariance matrix of the ensemble to be updated without the need for the covariances to be explicitly calculated which can be computationally expensive (Bishop et al., 2001; Hunt et al., 2007). These update equations are used to update the

error component only as will be discussed in Section 4.1.

## 4   Spatially consistent error-correction method for river discharge

In this section we describe how we use the data assimilation techniques discussed in Section 3, namely state augmentation and the LETKF, to correct the hindcasts across the domain including at ungauged locations. The correction is applied in a post-processing environment, avoiding the need for additional executions of the hydrological model which can be computationally

expensive. The proposed method consists of two steps: 1) updating the error ensemble (defined in Section 3.1), and 2) adjusting





the hindcast ensemble members using the updated error ensemble (Fig. 1). Specific experimental design choices are discussed in Section 5.



**Figure 1.** Schematic of the new error-correction method for gauged and ungauged locations. Coloured boxes indicate different components of the method. An initial error ensemble is created for timestep k=1 (green box). Then, the error ensemble is augmented to the hindcast ensemble (purple box). At each timestep the covariance of the augmented ensemble is inflated (cyan box) before being updated using the LETKF which uses localisation to improve the results of the update (collectively the orange box). The updated error ensemble is adjusted to ensure non-negative discharge values (light grey box) before being used to error-correct the hindcast (yellow box). The non-negative error ensemble is propagated to the next timestep (red arrows). More details are provided for each component in the section indicated in the top left corner of the corresponding box.

## 4.1 Updating the error ensemble

At each timestep the error ensemble is updated to estimate the optimal set of error vectors to correct the hindcast at that timestep. The update is performed using the LETKF defined in Section 3.2. Equations (14) and (15) are the Kalman update





equations for the augmented state. Using the definition of the augmented state (see Eq. (9)) the update equations for the error ensemble only are,

$$\overline{\mathbf{b}}_k^a = \overline{\mathbf{b}}_k^f + \mathbf{K}_{\mathbf{b}_k}(\mathbf{y}_k - \widehat{\mathbf{H}}_k \overline{\mathbf{w}}_k^f) \tag{16}$$

and

$$\mathbf{B}_k^a = \mathbf{B}_k^f \mathbf{T}_k. \tag{17}$$

As the hindcast component is not explicitly evolved (see Section 3.2), we assume that the raw hindcast is a good approximation for the hindcast analysis state were the component updated. This allows the substitution of the precomputed hindcast in place of the propagated state at the next timestep. Thus, the updated mean of the augmented ensemble can be defined as

$$\overline{\mathbf{w}}_k^a = \begin{pmatrix} \overline{\mathbf{x}}_k \\ \overline{\mathbf{b}}_k^a \end{pmatrix} \in \mathbb{R}^{2n}. \tag{18}$$

where $\overline{\mathbf{x}}_k$ is the ensemble mean of the raw hindcast ensemble and $\overline{\mathbf{b}}_k^a$ is the updated error ensemble mean (Eq. (16)). The perturbation matrix of the updated augmented ensemble follows a similar pattern such that

$$\mathbf{W}_k^a = \begin{pmatrix} \mathbf{X}_k \\ \mathbf{B}_k^a \end{pmatrix}. \tag{19}$$

where $\mathbf{X}_k$ is the ensemble perturbation matrix of the raw hindcast ensemble and $\mathbf{B}_k^a$ is the updated error ensemble perturbation matrix (Eq. (17)). The assumptions made in Eq. (18) and Eq. (19) make our system suboptimal. However, we provide proof-of-concept in this study that the resulting error ensemble improves the skill of the hindcast (see Section 7.1).

The Kalman filter is not constrained to enforce non-negativity of the analysis state, and therefore, could lead to negative discharge values for some grid boxes if the cross-covariances are incorrectly defined. We enforce non-negativity by further adjusting the error ensemble members after the LETKF update step (Fig 1). The adjustment is done separately for each grid box and each ensemble member only if they result in a negative river discharge as follows:

$$If \ \mathbf{x}_k^{(i)}[j] + \mathbf{b}_k^{(i)a}[j] < 0, \qquad\qquad then \ \hat{\mathbf{b}}_k^{(i)a}[j] = -\mathbf{x}_k^{(i)}[j] + ||\zeta_k|| \tag{20}$$

where $\hat{\mathbf{b}}_k^{(i)}$ is the adjusted error-ensemble member that results in non-negative discharge, $j$ indicates the $j$-th grid box, $i$ indicates the $i$-th ensemble member, $||.||$ indicates the modulus, and $\zeta_k$ a random noise value sampled from a Gaussian distribution with mean 0 and standard deviation equal to 10% of the standard deviation of the updated error ensemble at the grid-box of interest.

The updated positive-definite augmented states are propagated to the next timestep as defined in Eq. (12). The updated positive-definite augmented states are also used to error-correct the hindcast (Section 4.2).





## 4.2 Adjusting the forecast

After the error component of the augmented state has been updated using Eqs. (16) and (17), and non-negativity has been enforced (Section 4.1), the error ensemble members are added to the respective hindcast ensemble members such that

$$\mathbf{x}_k^{new,(i)} = \mathbf{x}_k^{(i)} + \hat{\mathbf{b}}_k^{(i)a} \tag{21}$$

where $\mathbf{x}_k^{new,(i)}$ and $\mathbf{x}_k^{(i)}$ are the $i$-th ensemble members of the error-corrected and raw hindcast ensembles, respectively, and $\hat{\mathbf{b}}_k^{(i)}$ is the error vector associated with the $i$-th error ensemble member where the caret indicates a non-negativity check has been applied. Consequently, the error-corrected hindcast ensemble mean and perturbation matrix are given by

$$\overline{\mathbf{x}}_k^{new} = \overline{\mathbf{x}}_k + \overline{\hat{\mathbf{b}}}_k^a \tag{22}$$

and

$$\mathbf{X}_k^{new} = \mathbf{X}_k + \hat{\mathbf{B}}_k^a. \tag{23}$$

The additive spread correction matrix defined in Eq (6) is the result of calculating the covariance matrix of the error-corrected hindcast ensemble as in Eq. (3). The form of the additive spread correction matrix is

$$\mathbf{\Gamma}_k = \mathbf{X}_k\hat{\mathbf{B}}_k^{aT} + \hat{\mathbf{B}}_k^a\mathbf{X}_k^T + \hat{\mathbf{B}}_k^a\hat{\mathbf{B}}_k^{aT} \tag{24}$$

where $\mathbf{X}_k$ and $\hat{\mathbf{B}}_k^a$ are the perturbation matrices of the raw hindcast and error ensembles, respectively, and the superscript $T$ indicates the matrix transpose (Section 5.2 in Martin, 2001).

## 5 Experimental implementation

In Section 4 we presented a new method of spreading observation information to ungauged locations in a post-processing environment based on common data assimilation techniques. In this section, we describe three key components of the method—localisation,

covariance inflation, and the generation of the initial error ensemble—which are crucial for its performance but can be implemented in various ways.

### 5.1 Localisation

Localisation is used to reduce the effect of spurious correlations which can arise due to sampling errors caused by the small ensemble size (Hamill et al., 2001; Hunt et al., 2007). The LETKF uses observation localisation which reduces the impact of

255 observations by multiplying the inverse of the observation-error covariance matrix by a *localisation matrix*, $\boldsymbol{\rho} \in \mathbb{R}^{p_k \times p_k}$, such that

$$\mathbf{R}^{-1} = \boldsymbol{\rho} \circ \mathbf{R}_{nl}^{-1} \tag{25}$$



where $\mathbf{R} \in \mathbb{R}^{p_k \times p_k}$ is the localised observation-error covariance matrix used in the LETKF (Section 3.2), $\mathbf{R}_{nl} \in \mathbb{R}^{p_k \times p_k}$ is the non-localised observation-error covariance matrix, and the symbol ∘ indicates the Schur product (also known as the Hadamard product) which is an element-wise matrix multiplication (Golub and Van Loan, 2013). We assume that $\mathbf{R}_{nl}$ and, by definition, $\mathbf{R}$ are diagonal matrices. In this study we use distance-based localisation so the impact of the multiplication described in Eq. (25) is to increase the effective uncertainty of distant observations and thus decrease their impact on the analysis state. The impact of the localisation on the spatial extent of the analysis increments is demonstrated in Section 7.1.

The localisation matrix is defined using the Gaspari-Cohn function which has a parameter called the *localisation length scale* (Eq. 4.10 in Gaspari and Cohn (1999)). The Gaspari-Cohn function smoothly decreases the weights assigned to an observation as the distance from the observation location increases, starting from a value of 1 at the observation location and reaching 0 for distances greater than twice the localization length scale (pink box, Fig. 1). In this study, the distance is calculated along the river network which has been shown to improve the analysis for fluvial applications (García-Pintado et al., 2015; El Gharamti et al., 2021; Khaniya et al., 2022). The distance between a grid-box and the location of an observation is calculated using the local drainage direction map and the channel length used in the hydrological model (Choulga et al., 2023). As the distance is defined along the river network, observations cannot impact grid-boxes in a different drainage basin.

The localisation length scale is often a tuned parameter but the tuning process can be time and resource intensive. We propose instead for the localisation length scale to be defined as the maximum distance between any grid point and its closest observation. This 1) ensures that all grid boxes are updated in the update step of the LETKF reducing the potential for discontinuities in the analysis state, 2) can adapt to changes in the availability of observations, and 3) can be applied to different domains and hydrological model configurations without requiring a tuning experiment.

## 5.2  Covariance inflation

Small ensemble sizes can cause underestimation of the ensemble spread which in turn reduces the impact of the observations on the analysis (Furrer and Bengtsson, 2007). In addition to the issues caused by the small ensemble size, we also make the simplified assumption that the error ensemble is constant between timesteps (Eq. (11)) which could introduce model errors into the ensemble (Evensen et al., 2022). Covariance inflation is an approach often used to ameliorate these issues, although an inflation method that is optimal for all situations has yet to be identified (Duc et al., 2020; Scheffler et al., 2022).

We aim to inflate the ensemble perturbation matrix such that at time $k+1$ the spread better represents the true uncertainty of the mean error prior to the update step. Due to the unusual approach of using predefined ensembles, we propose a new method to inflate the covariance of the error ensemble. We take inspiration from the 'relaxation to prior perturbations' technique (RTPP, Zhang et al., 2004; Kotsuki et al., 2017) which blends the analysis perturbation matrix with the perturbation matrix prior to the analysis step. This results in both additive and multiplicative inflation that is proportional to the impact of the assimilation of observations (Whitaker and Hamill, 2012). Rather than accounting for errors introduced in the update step we want to account for errors introduced primarily in the propagation step. We therefore adapt the RTPP method to blend the propagated analysis perturbation matrix, $\mathbf{W}_k^a$ defined in Eq. (19), and an alternative estimate of the perturbation matrix at the next timesteps, $\mathbf{W}_{k+1}^{est}$, such that





$$\mathbf{W}_{k+1}^{inf} = (1-\alpha) \begin{pmatrix} \mathbf{M}_k & \mathbf{I}_k^{n \times n} \\ \mathbf{0}_k & \mathbf{I}_k^{n \times n} \end{pmatrix} \mathbf{W}_k^a + \alpha \mathbf{W}_{k+1}^{est} \tag{26}$$

where $\mathbf{M}_k$ and $\mathbf{I}_k^{n \times n}$ are evolution matrices introduced in Section 3.1 and $\alpha$ is an inflation parameter that must be defined. The estimation of $\mathbf{W}_{k+1}^{est}$ could use an alternative model to evolve the analysis perturbation matrix forward or be a climatological matrix (Valler et al., 2019). In this study we give $\mathbf{W}_{k+1}^{est}$ the form

$$\mathbf{W}_{k+1}^{est} = \begin{pmatrix} \mathbf{X}_{k+1}^{est} + \mathbf{B}_{k+1}^{est} \\ \mathbf{B}_{k+1}^{est} \end{pmatrix}. \tag{27}$$

where $\mathbf{X}_{k+1}^{est}$ and $\mathbf{B}_{k+1}^{est}$ can be estimated separately. We assume the hindcast covariance matrix is correct such that $\mathbf{X}_{k+1}^{est} = \mathbf{X}_{k+1}$, and estimate the error covariance matrix at time $k+1$ as proportional to the state covariance matrix (Dee, 2005; Martin et al., 2002). We assume that the constant of proportionality is 1 such that $\mathbf{B}_{k+1}^{est} = \mathbf{X}_{k+1}$. When substituted into Eq. (26), this form of $\mathbf{W}_{k+1}^{est}$ maintains consistency between the error terms in the hindcast and error components of the augmented state.

The inflation parameter, $\alpha_k$, controls the weighting of propagated analysis perturbation matrix and the estimated perturbation matrix. Here, $\alpha_k$ determines how much of the uncertainty at time $k+1$ is due to uncertainty at time $k$ and how much is not captured by the propagated matrix. We assume that the change in the variance of the hindcast ensemble between timesteps is an indication of how much the spread of the error ensemble would change if the propagation model of the error was correct. Therefore, we define the inflation parameter as the fractional change in the hindcast variance,

$$\alpha_k = \frac{1}{k} \sum_{l=k-2}^{l=k} max\left\{ \frac{|Tr(\mathbf{P}_l) - Tr(\mathbf{P}_{l+1})|}{Tr(\mathbf{P}_l)}, 1 \right\} \tag{28}$$

where $k$ is the the current timesteps and $Tr(\mathbf{P}_l)$ is the trace of the raw hindcast covariance matrix at timesteps $l$. A maximum value of 1 is set to avoid instabilities, particularly at short lead-times where the change in variance between timesteps can be large. The average over the past three timesteps is taken to ensure that alpha is smoothly changing between timesteps, again to avoid instabilities. An inflation value of 1 suggests the uncertainty of the modelled state has changed so much between timesteps that the uncertainty at the previous timestep is no longer relevant. An inflation values of 0 implies the uncertainty at the previous timesteps should be trusted.

### 5.3 Initial error ensemble

We must define an initial error ensemble to perform the state augmentation at the first timestep. Due to the application to a post-processing environment there is no "warm-up" period in which a state of equilibrium can be reached, and therefore the initial error must be physically plausible. Here, the initial error ensemble is defined using three sets of river discharge data: in-situ observations, $\mathbf{y}_k \in \mathbb{R}^{p_k}$, simulations created by forcing a hydrological model with meteorological observations, $\mathbf{s}_k \in \mathbb{R}^n$, and the ensemble mean and ensemble perturbation matrix of a single lead-time from a previous hindcast, $\overline{\mathbf{x}}_k \in \mathbb{R}^n$





and $\mathbf{X}_k \in \mathbb{R}^{n \times N}$. A single ensemble is generated for the full EFAS domain and then the elements associated with the domain

of interest (in this study the Rhine-Meuse catchment) are extracted.

We define the initial error ensemble in two steps: 1) the error mean is estimated based on the errors due to the hydrological model, and 2) ensemble perturbations are estimated based on the perturbations of hindcast members. The mean of the initial error ensemble is estimated as follows:

1. **Calculate the errors at gauged locations:** The average relative error of the simulation compared to the observations

over the past $d$ days (here 10 days) at all $p_k$ stations with available observations, $\boldsymbol{\delta} \in \mathbb{R}^{p_k}$ is calculated as

$$\boldsymbol{\delta}[j] = \sum_{k=-d}^{k=-1} \frac{\mathbf{y}_k[j] - \mathbf{s}_k[j]}{\mathbf{s}_k[j]} \tag{29}$$

where $\boldsymbol{\delta}[j]$, $\mathbf{y}_k[j]$ and $\mathbf{s}_k[j]$ are the relative error, the observation and the simulated value at station $j$ at time $k$, respectively. If the value of $\boldsymbol{\delta}[j]$ is greater than 1 (or less than -1) then $\boldsymbol{\delta}[j]$ is set to 1 (or -1). This reduces the impact of representation errors due to the mapping of stations, for example.

2. **Interpolate the errors to ungauged locations:** Inverse distance weighted interpolation is used to estimate the average relative error at ungauged locations ensuring closer stations have a greater influence (Lu and Wong, 2008). The Euclidean distance, denoted $d_{gj}$, is calculated between a grid-box, $g$, and each of the closest $G$ stations (here 100 stations), and the average relative error weighted accordingly. The Euclidean distance is used here to allow the method to be applied to all catchments. Therefore, the inverse distance weighted formula used to calculate the relative error at grid-box $g$, denoted

$\boldsymbol{\Delta}[g]$, is

$$\boldsymbol{\Delta}[g] = \frac{\sum_{j=1}^{100} \boldsymbol{\delta}[j]/\sqrt{d_{gj}}}{\sum_{j=1}^{100} 1/\sqrt{d_{gj}}} \tag{30}$$

3. **Impose the river network structure:** The mean of the initial error ensemble is calculated by multiplying the field of estimated relative errors, $\boldsymbol{\Delta}$, with the simulation at time $t = -1$, $\mathbf{s}_{-1}$, such that at grid-box $g$ the initial error ensemble mean, $\overline{\mathbf{b}}_1^f[g]$ is calculated as

$$\overline{\mathbf{b}}_1^f[g] = \boldsymbol{\Delta}[g] \times \mathbf{s}_{-1} \tag{31}$$

where the superscript $f$ indicates the ensemble has not been updated by the LETKF (Section 3.2).This enforces the spatial pattern of the river network by ensuring the value of the initial error mean is proportional to the magnitude of the discharge in the river.

The perturbations from the ensemble mean are then defined as follows. Since the form of the covariance matrix of the errors

is unknown, we use a common technique of scaling the system state covariance matrix (e.g. Martin et al., 2002). We use the ensemble members from the second lead-time of the hindcast from two days prior. This lead-time was selected as the spread of the ensemble at a lead-time of one day can often be very narrow due to the use of a single set of initial conditions (see 6.1). The steps to define the initial error perturbations are:





1. **Calculate the ensemble statistics:** The ensemble mean and perturbation matrix of the the second lead-time of the hindcast from two days prior are calculated. The valid-time for these ensemble values is, $t = 0$ so they are denoted $\overline{\mathbf{x}}_0$ and $\mathbf{X}_0$, respectively.

2. **Inflate the covariance matrix:** The spread of the hindcast ensemble should account for the uncertainty due to the meteorological forcings. To ensure the variance is correct we scale the perturbation matrix by the error of the ensemble mean compared to the simulation forced by meteorological observations. A vector of scaling factors, $f$ is defined such that

$$\mathbf{f} = \left( \frac{\mathbf{s}_0[1] - \overline{\mathbf{x}}_0[1]}{\overline{\mathbf{x}}_0[1]}, \ \frac{\mathbf{s}_0[2] - \overline{\mathbf{x}}_0[2]}{\overline{\mathbf{x}}_0[2]}, \ldots, \ \frac{\mathbf{s}_0[n] - \overline{\mathbf{x}}_0[n]}{\overline{\mathbf{x}}_0[n]} \right)^T \tag{32}$$

where $\mathbf{s}_0[n]$ is the simulation at the $n$-th gridbox. These scaling factors can be very small so we set a requirement that the scaled standard deviation at each grid-box must not go below 10% of the simulated value, $\mathbf{s}_0[j]/10$. In practice this is done in two steps: i) scale the perturbation matrix by the vector $\mathbf{f}$ such that

$$\mathbf{B}_1^f = \mathbf{f}\mathbf{X}_0 \tag{33}$$

and then ii) reinflate the spread where necessary. By scaling the perturbation matrix in this way we relate the spatial variability of the ensemble spread to the error due to the meteorological forcings.

The error mean, $\overline{\mathbf{b}}_1^f$, and the error perturbation matrix, $\mathbf{B}_1^f$ are used to define the error ensemble at timesteps $k = 1$ where they are updated using the LETKF with state augmentation as described in Section 4.1.

# 6 Evaluation strategy

## 6.1 European Flood Awareness System (EFAS)

The hindcasts used in this study were produced by the European Flood Awareness System (EFAS) as operational forecasts (Barnard et al., 2020). EFAS is part of the Early Warning component of the European Commission's Copernicus Emergency Management Service (CEMS), and aims to provide complementary forecast information to hydro-meteorological services throughout Europe (Matthews et al., 2025). EFAS streamflow forecasts are produced by forcing a calibrated hydrological model, LISFLOOD (De Roo et al., 2000; Van Der Knijff et al., 2010; Arnal et al., 2019), with the output from meteorological numerical weather prediction (NWP) systems. Whilst the operational EFAS system is a multi-model system with four sets of meteorological forcings, in this study we focus only on the medium-range river discharge forecasts generated with meteorological forcings from the 51-member medium-range ensemble from the European Center for Medium-range Weather Forecasts (ECMWF) due to its large ensemble size. The meteorological forcings are interpolated to the EFAS grid. A single set of initial hydrological conditions are used for all ensemble members often leading to small ensemble spreads at short lead-times. The spread then increases as the different meteorological forcings propagate through the system. No data assimilation is performed





in the generation of the initial hydrological conditions. Instead, the LISFLOOD hydrological model is forced with meteorological observations (and meteorological forecasts when observations are not available) to generate the initial conditions (Smith
et al., 2016).

As an operational system, EFAS is constantly evolving. For the evaluation presented here we use EFAS version 4 (operational from 14 October 2020 to 20 September 2023) aggregated to daily timesteps with a maximum lead-time of 15 days. The ensembles have 51 members and predict the average river discharge for each timestep for each grid-box within the domain (see 6.2). The hindcasts have a spatial resolution of 5km $\times$ 5km with a ETRS89 Lambert Azimuthal Equal Area Coordinate
Reference System. Hindcast from the 00 UTC daily cycle are used resulting in a total of 365 hindcasts used in the evaluation.

## 6.2    Rhine-Meuse catchment

The Rhine-Meuse catchment has a drainage area 195,300 $km^2$, a channel length of about 38,370 $km$ in EFAS, and consists of 7812 grid-boxes. It is the 5th largest catchment in EFAS. The Rhine river originates in the Swiss Alps, flows through the Central Uplands and the North European Plain, before finally discharging into the North Sea. The Meuse river originates from
the Langres Plateau in France, flows through the Ardennes Massif and the low-lying plains of the Netherlands, before merging with the Rhine and entering the North Sea. The catchment consists of rivers of different sizes, topologies, and levels of human influence, making it an ideal test catchment to see how the method deals with changes along the river network.

## 6.3    Observations

The Rhine-Meuse catchment has a dense river gauging station network. The main set of observations used in this study are
daily river discharge observations from 89 stations across the Rhine-Meuse catchment for the time period from 21 December 2020 to 15 January 2022. The minimum value across the stations is 0.516 $m^3 s^{-1}$ and the maximum value is 7662.917 $m^3 s^{-1}$. These observations were assimilated as part of the error-correction method to update error ensemble and used in the evaluation of the corrected forecasts (Section 6.4 describes the cross-validation approach used). Whilst the error-correction method can adapt to missing observations, these 89 stations were selected as they have no missing data for the time period of interest
allowing this analysis to focus on the spread of observational information to ungauged locations. The maximum distance between any grid-box and the closest of the 89 stations is 262 $km$ which is set as our localisation length scale (cut-off distance is therefore 524 $km$; see Section 5.1). In addition to these stations, all available observations from across Europe, were used to generate the initial error ensembles (total 505 stations). All river discharge observations were provided by local and national authorities and collated by the CEMS Hydrological Data Collection Centre (see https://confluence.ecmwf.int/display/CEMS/
EFAS+contributors).

The construction of the non-localised observation error covariance matrix, $\mathbf{R}_k^{nl}$, is a key component of all data assimilation methods. The matrix describes the uncertainty associated with each observation and describes the correlation between errors of different observations (Stewart et al., 2013; Fowler et al., 2018). In this study, we assume that the observation errors from different gauge stations are uncorrelated such that $\mathbf{R}_k^{nl}$ is a diagonal matrix with all off-diagonal elements set to 0. We also





assume that the standard deviation of the observation errors is 10% of the observation magnitude (Refsgaard et al., 2006; McMillan et al., 2018, 2012).

In the leave-one-out verification experiments (see Section 6.4) we use the observations from the non-assimilated station as validation data and assume they are the truth with no errors.

## 6.4    Experiments

We use three experimental schemes to investigate the effect that the error-correction scheme has on the ensemble hindcasts.

1. **Single station experiments:** Only observations from one of the 89 station are assimilated when estimating the error-vector. All available observations are used in the generation of the initial error ensemble. These experiments allow the impact of an observation to be identified and allow the effects of localisation to be explored.

2. **All station experiments:** Observations from all stations are assimilated when estimating the error-vector and used in
the generation of the initial error ensemble. These experiments allow the complete method to be assessed and for any spatiotemporal inconsistencies to be identified.

3. **Leave-one-out experiments:** Observations are withheld from one of the 89 stations and are not assimilated when estimating the error-vector nor used in the generation of the initial error ensemble. This cross-validation framework allows the skill of the adjusted hindcasts to be assessed at the locations of stations as if they were ungauged locations.

Each experiment scheme is applied to all hindcasts from 1 January 2021 to 31 December 2021. However, for brevity, for the single station and all station experiments we only discuss two hindcasts: 7 July 2021 and 8 October 2021. These dates represent high and normal flow conditions, respectively, allowing the ability of the method to be assessed for different circumstances.

## 6.5    Evaluation metrics

The following metrics are used to investigate the skill of the error-corrected hindcast ensemble mean and the reliability of the
ensemble spread.

For the ensemble mean, the three components of the modified Kling-Gupta Efficiency: correlation, mean bias, and variability bias are used to assess different types of errors within the ensemble mean (Kling et al. (2012); Gupta et al. (2009)). Pearson's correlation coefficient measures the linear relationship between the simulated timeseries and the observations indicating timing errors (score range $[-1, 1]$). The mean bias given by the ratio between the mean of the simulated timeseries and mean of the
observations indicates whether the flow is consistently over or under-estimated (score range $(-\infty, +\infty)$). The variability bias given by the ratio between the coefficient of variation of the simulation and the coefficient of variation of the observations indicates whether the variability in the flow is consistently over or under-estimated (score range $(-\infty, +\infty)$). All three components have a perfect score of 1. Additionally, to investigate whether the magnitude of the error of the forecast mean is decreased by the proposed method we use the Normalised Mean Absolute Error (NMAE Hodson, 2022; Jackson et al., 2019). The metric is





normalised by dividing by the mean of the observations for that station. Normalising the metric makes the scores at different stations comparable. The NMAE has a perfect score of 0.

To analyse the reliability of the spread of the ensemble forecast we use the rank histogram (Harrison et al., 1995; Anderson, 1996; Hamill and Colucci, 1997; Talagrand, 1999). To generate the histogram the rank of the observation relative to the sorted ensemble values is calculated for each hindcast. The frequencies with which the observation has a rank from 1 to M + 1 are

plotted as a histogram. The shape of the histogram provides information about the reliability of the ensemble spread and bias of the ensemble (Hamill, 2001).

## 7 Results

### 7.1 Impact of assimilating observations

In this section we investigate the spatial and lead-time dependent impact of assimilating the observations. To assess the spatial

impact of these observations, we analyze the analysis increments of the mean — the difference between the ensemble mean before and after the update step (term 2 in Eq. (16)) — across the domain at a single lead-time for single station experiments and all station experiments (Fig. 2). We focus on single station experiments for the Bonn station on the Rhine (left panels) and the Uckange station on the Moselle (middle panels) for hindcasts generated on 8 October 2021 (upper panels) and 7 July 2021 (lower panels), which represent normal and high flow scenarios, respectively.

For 8 October 2021, the assimilation of an observation at Bonn results in the largest analysis increments near the observation location, with the increments diminishing to zero at distances greater than 524 km due to localization (background color of Fig. 2a). Increasing (decreasing) the localisation length scale results in a more (less) gradual dampening of the analysis increments and more (fewer) grid-boxes being impacted by a single observation (not shown). The number of grid-boxes in the localization regions of the Bonn and Uckange stations differ (4662 grid boxes and 2451 grid boxes, respectively) because the

distance is calculated along the river network and the channel length within each grid box is not constant (e.g., Figs. 2a and 2b). Interestingly, in the Uckange experiment, the largest increments occur not near the station, but along the Rhine near the confluence with the Moselle (Fig. 2b). In both experiments, the increments tend to be larger along bigger rivers, with smaller rivers showing smaller increments. This occurs due to large ensemble covariances between the location of the assimilated observation and locations along the bigger rivers (Fig. 3).

The spreading of observational information along the river network is dictated by the cross-covariances between the error component and the hindcast component of the augmented ensemble prior to the update step (Section 3.2). The magnitude of the cross-covariance between two locations depends on the correlation at the two locations, the variance of the augmented ensemble at both locations, and the dampening enforced by localisation. The correlation between the location of the Uckange station and a grid-box is highest along the same river stretch (the Moselle) and decreases at longer distances from the station

(Fig. 3a). Nearby grid boxes that are not on the same river stretch have lower correlations in general. Comparing Figs. 3a and 3d indicates that along-the-river localisation is appropriate for this application, as with other hydrological data assimilation



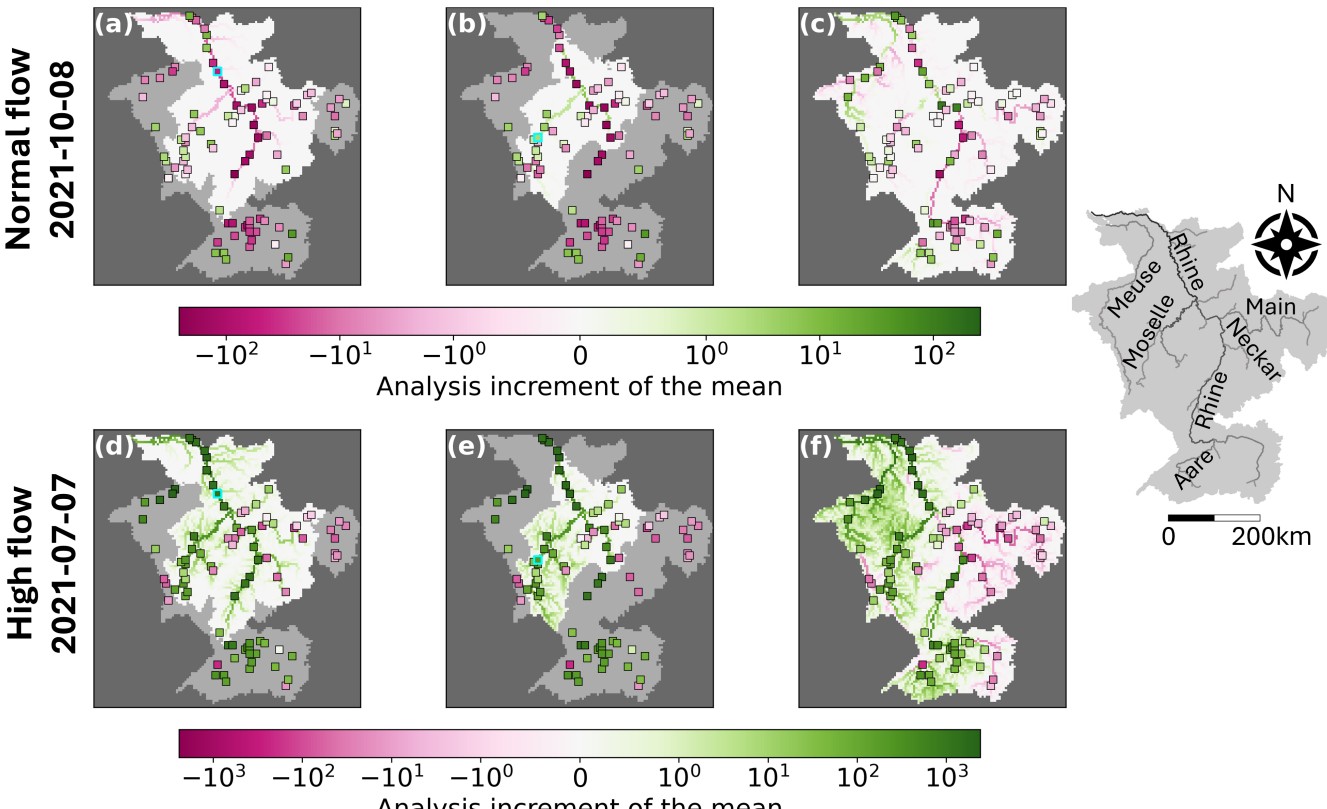

**Figure 2.** Analysis increments of the mean for a lead-time of 9 days for single station (a, b, d, and e) and all station (c and f) experiments for the hindcasts generated on 8 October 2021 (upper panels) and 7 July 2021 (lower panels). Assimilated stations for the single station experiments (cyan outline) are the Bonn station on the Rhine (a and d) and the Uckange station on the Moselle (b and e). The shaded region of the catchment is outside the localisation length of the assimilated station. Markers show the innovation at all stations. Catchment area: 195,300 $km^2$.

systems (e.g., El Gharamti et al., 2021), as it results in the impact of an observation being restricted to locations with higher, physically plausible, correlations.

The magnitude of the cross-covariances are similar along the Moselle and for some parts of the Rhine despite lower correlations and the application of localisation (Fig. 3b and 3e). Whilst the correlation initially decreases with increasing distance, the magnitude of the non-localised cross-covariances is primarily dependent on the size of the river (note the horizontal bands of Strahler orders (a measure of stream size where larger orders indicate larger rivers Strahler, 1957) in Fig. 3f). Localisation enforces a dependence on distance such that smaller rivers near the station have a larger localised cross-covariance than large rivers very far from the station (Fig. 3e). However, some grid-boxes on the Rhine (Strahler order of 6) still have larger

cross-covariances than smaller rivers that are closer to the station (Fig. 3e). The impact of this can be seen in Fig. 2b where the analysis increments along the Rhine are larger than those along parts of the Moselle. Note that the similarity between the



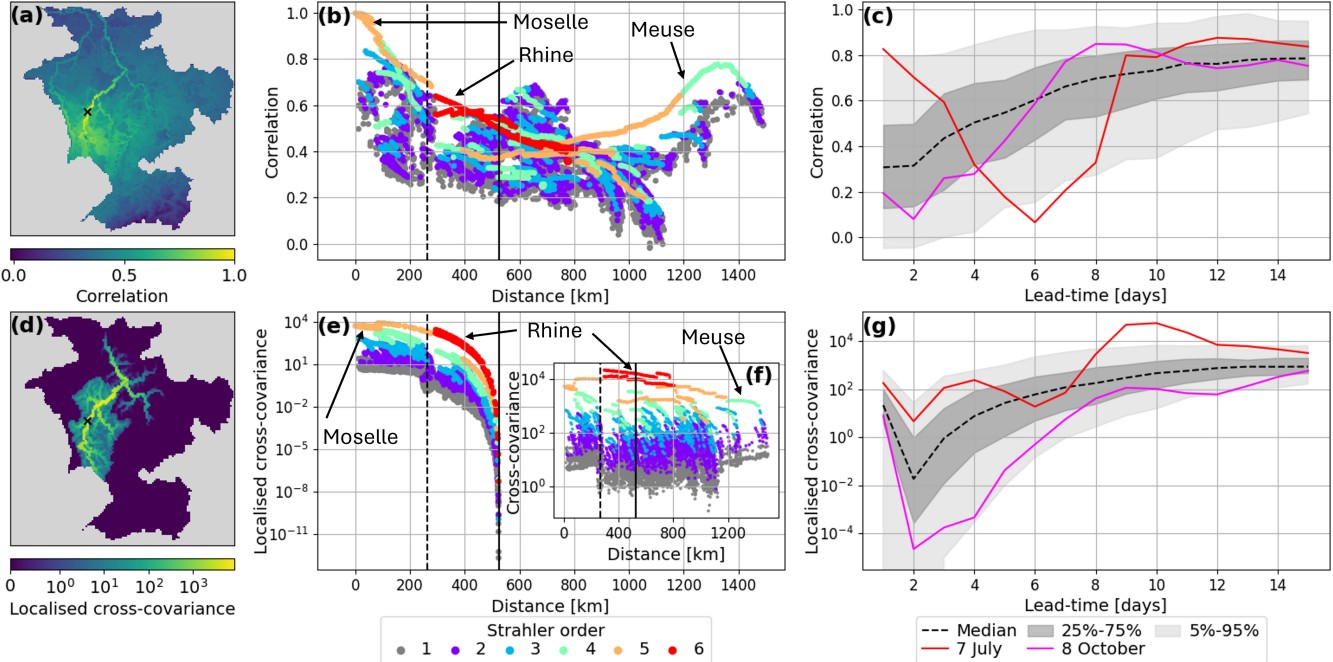

**Figure 3.** Ensemble correlations (upper panels) and cross-covariances (lower panels) between the error component and the hindcast component of the augmented state prior to the update step for all station experiments. Map of the correlation (a) and localised cross-covariance (d) averaged across all lead-times and forecasts between the Uckange station (shown by black cross) and all grid-boxes. Scatter plot of the correlation (b), localised cross-covariance (e), and non-localised cross-covariance (f) averaged across all lead-times and forecasts between the Uckange station and all grid boxes against distance from the Uckange station. Grid-boxes on rivers discussed in the text are broadly indicated by the arrows. Dashed black line shows the localisation length scale and the solid black line shows the effective cut-off point beyond which the observation has no impact (twice the localisation length scale; see Section 5.1). Correlation (c) and localised cross-covariance (g) between the Uckange station on the Moselle and the Bonn station on the Rhine for all forecast and for each lead-time of the hindcast (365 values per lead-time, one for each forecast).

localisation length scale (dashed line) and the distance between the Uckange station and grid-boxes on the Rhine (change from a Strahler order of 5 to 6) is coincidental but does suggest that the method for defining the localisation length scale (see Section 5.1) is capable of capturing the order of magnitude of the relevant spatial scales for the Rhine catchment.

In Fig. 2, the square markers indicate the innovation — the difference between the observation and the error-corrected ensemble mean prior to the update step. Ideally, the analysis increment (background colour in Fig. 2) should reflect similar behavior to the innovations within the localisation region, implying that the ensemble is being adjusted towards the observations at each station. For 8 October, at Bonn the innovation is negative and results in negative analysis increments across the domain (Fig. 2a). For the Uckange station the innovation is positive and the analysis increments are also all positive (Fig. 2b) indicating

positive ensemble covariances (Fig. 3d). For both of the 8 October experiments the analysis increments match the sign of the





innovation vectors for neighbouring stations (Figs. 2a and 2b). At greater distances the analysis increments do not follow the same behaviour as the innovations. For example, the innovations along the Rhine in the Uckange experiment are negative whilst the analysis increments are positive (Fig. 2b) suggesting spurious ensemble covariances between the Uckange station and locations on the Rhine. In the experiments for 7 July, the analysis increments show a similar behaviour to the innovations

for a much greater distance along the river network (Figs. 2d and 2e). In contrast to the 8 October experiments (Fig. 2c), the innovations for the 7 July experiments are more spatially homogeneous indicating a greater spatial correlation length. This is likely due to the low-pressure system which covered large parts of the West of the catchment during the hindcast period of the 7 July experiments (Mohr et al., 2023). In Figure 3b, the average correlation can be seen to begin to increase again for larger distances. This is due to grid boxes on different rivers (here, the Meuse and the Moselle; see rivers names in Fig. 2) being close

geographically but far apart along the river network. Geographically close locations may be impacted by the same weather systems even if the water drains into different rivers as happened for the 7 July period.

Whilst the spatial heterogeneity for 8 October suggests that the assimilation of an observation from a single station cannot correct the entire domain (Figs. 2a and 2b), when all observations are assimilated the analysis increments vary across the domain (Fig. 2c). This heterogeneity of the analysis shows the ability of the method to vary the correction across the domain

adapting to the errors in different stretches of the river. The analysis increments are smoothly changing along a river stretch therefore, the changes to the error-corrected ensemble will also be smoothly changing spatially.

Another difference between the 8 October and the 7 July experiments is that for the 7 July hindcast small rivers exhibit larger increments, indicating a greater impact from the assimilated observation (Figs. 2c and 2f). Two factors contribute to this increased influence. First, the increased spatial correlation length means the observation is more informative for longer

distances. However, the correlation between, for example, the locations of the Uckange and Bonn stations is higher at a lead-time of 9 days for the 8 October than for the 7 July (Fig. 3c). Therefore, the second factor, larger ensemble variances in the 7 July period compared to the 8 October period, is likely the more dominant component (Figs. 4b, 4c, 4e, and 4f). The increase in spread increases the cross-covariances (Fig. 3g) and allows the observation to have more influence.

Figure 4 shows the trajectories of the three ensembles used in the LETKF for the 7 July hindcast for a single station

experiment where observations are assimilated at the Uckange station: the raw hindcast (left columns), the hindcast component of the augmented ensemble (middle columns), and the error component of the augmented ensemble (right columns). The evolution of the augmented ensemble are discussed in Sections 3.1-5. The lower panels show the trajectories at the Bonn stations for which no observations are assimilated during this experiment. By plotting the raw hindcast trajectories and the observations we can visualise the errors to be estimated. We can see that for both stations the error of the hindcast mean is

negative (observations are smaller than the hindcast mean) for lead-times up to 8 days, and positive at longer lead-times. Whilst this behaviour is similar for the Bonn station, the magnitude of the error is different by a factor of 10 at most lead-times.

The middle column shows the hindcast component of the augmented ensemble. We can see that using the raw hindcast as an approximation of the analysis state is not optimal. For example, at lead-times greater than 10 days at the Uckange station the update takes the ensemble further away from the observations (Fig. 4b). This occurs also at the Bonn station (Fig. 4e). This

is not unexpected as our approximation assumes the raw hindcast is more accurate than the hindcast corrected with the error





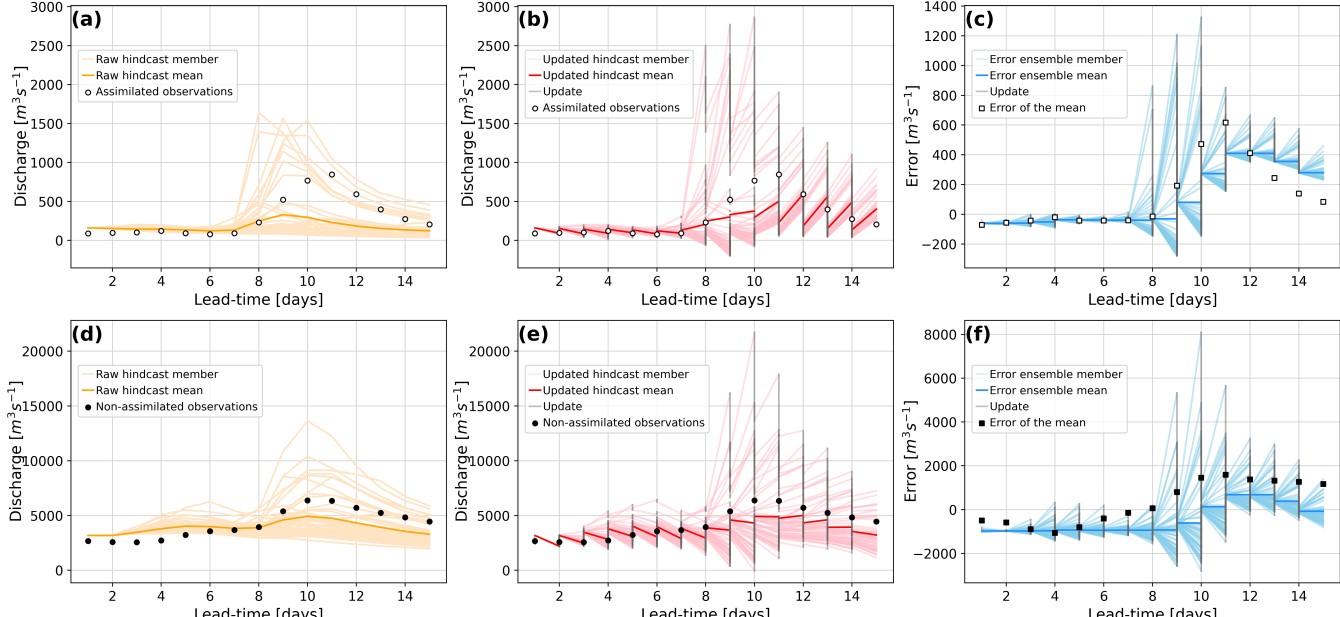

**Figure 4.** Ensemble trajectories for a single station experiment for the hindcast generated on 7 July 2021 for the assimilated station (Uckange station on the Moselle; panels a-c) and a non-assimilated station (Bonn station on the Rhine; panels d-f). The plots show the trajectory of all members and the ensemble mean of the raw hindcast ensemble (left panels), the hindcast component of the augmented state (middle panels), and the error ensemble members (right panels; different y-axis scale). Markers show the river discharge observations (a, b, d, and e), and the error of the raw hindcast mean (c and f).

ensemble members from the previous time step (Section 4.1). However, this assumption is necessary to propagate the hindcast to the next time step without the use of a hydrogical model (Section 3.1).

It is the error ensemble that is most important to our application (Figs. 4c and 4f). Despite the non-optimal formation of the analysis augmented state, the error ensembles are updated beneficially, with the analysis error ensemble mean moving closer to

the error of the raw hindcast mean at each lead-time for the assimilated location (Fig. 4c) and the non-assimilated location (Fig. 4f). At short lead-times the updates to the error ensemble at the Bonn station do not appear to be beneficial (Fig. 4f). However, as this experiment only assimilates observations from one station this discussion should be considered a demonstration of how the method updates proxy-ungauged locations rather than an evaluation of the error-corrected ensemble (which is provided in Section 7.2). First we note, that the updates at the assimilated location do not result in the error ensemble mean (dark

blue line) matching the error of the mean (markers). This is expected and is due to the consideration of the observational uncertainty within the LETKF. This ensures spatial consistency across assimilated and non-assimilated locations, and combines the modelled and observed data to estimate the true state of the system across the domain.

The error-ensemble is narrow after the update step and it is the covariance inflation that increases the spread between timesteps. The spread of the hindcast is due to meteorological forcings, predominantly precipitation. Therefore, in general,





the hindcast spread is larger for longer lead-times as the precipitation forecasts become more uncertain and this uncertainty is propagated along the river network, and higher river discharge values (when precipitation is above 0 mm). Since the covariance inflation technique presented here results in the blending of the hindcast perturbation matrix with the error-ensemble from the previous timestep, this behaviour in the hindcast spread is transferred to the error-ensemble. As demonstrated in Figs. 4c and 4f, this can result in the error ensemble spread being large for the rising limb of an event and smaller for the falling limb. This

can result in the error not being updated sufficiently, as seen after the peak in Fig. 4c and discussed later along with Fig. 5b.

## 7.2   Ensemble skill

In this section we look at whether the updated ensemble is more skillful than the raw hindcast ensemble. Using leave-one-out experiments we evaluate the ensemble mean and ensemble spread at proxy-ungauged locations (Section 6.4). The hydrographs in Fig. 5 show the raw and error-corrected ensembles for three proxy-ungauged locations from the leave-one-out experiments.

The hydrographs are used to illustrate the method's ability to correct the ensemble and some of the limitations.

### 7.2.1   Skill of the ensemble mean

To investigate the impact on different types of errors in the ensemble mean we calculate the correlation, mean bias, variability bias and the NMAE for each station and each lead-time (Section 6.5). Figure 6 compares the skill of the ensemble mean of the raw and the error-corrected ensembles focusing on the overall change in skill (a, d, g, and i), the spatial dependency of the skill

(b, e, h, and k), and the lead-time dependency of the skill (c, f, i, and l).

    The error-corrected ensemble means show a stronger correlation with observations than the raw hindcast ensemble means, with an average increase from 0.82 to 0.92, and an overall shift towards the perfect value of 1 (Fig. 6a). Figure 5a shows an example of how the error-corrected ensemble can better capture the dynamics of the observations improving the correlation. It can be seen that the resulting ensemble is temporally consistent and does not have improbable changes between timesteps.

However, at four stations the correlation worsens compared to the raw hindcast ensemble (Fig. 6b). The two most southern of these worsened stations, are near to confluences with larger rivers which have different correlation patterns in the raw hindcast to those of the two stations of interest (note the much lighter colours for nearby stations; Fig. 6a). The ensemble covariances are not capturing this change in regime correctly so the observational information is not being advantageously spread between these rivers. The remaining two stations are the most upstream stations on their rivers. At these, stations the updates made to the

error-corrected ensemble are dependent on observations assimilated downstream. The assimilated observations are therefore providing information about a past state of the river upstream which could be the cause of the decreased correlation (a measure of timing errors) at these upstream stations. Whilst most upstream stations are improved by the error-correction method, stations which have much smaller upstream areas that their closest downstream station tend to be improved less than those that have a similar upstream area, particularly if the distance to the neighboring station is large.

The error-corrected ensemble generally has a lower mean bias than the raw hindcast ensemble, with the average mean bias shifting from 1.027 (overestimation) to 1.004 (less overestimation). However, there is a slight shift towards underestimation (Fig. 6d). Just over half of the stations (47) show improvement in mean bias averaged across all lead times (Fig. 6e), but no






**Figure 5.** Raw and error-corrected hydrographs for proxy-ungauged locations in leave-one-out experiments at the Rees station on the Rhine (upstream area: 159,320 $km^3$) and the Mainleus station on the Main (upstream area: 1,164 $km^3$). Catchment illustrations indicate the location of the station (see Fig. 2 for rivernames and scale).





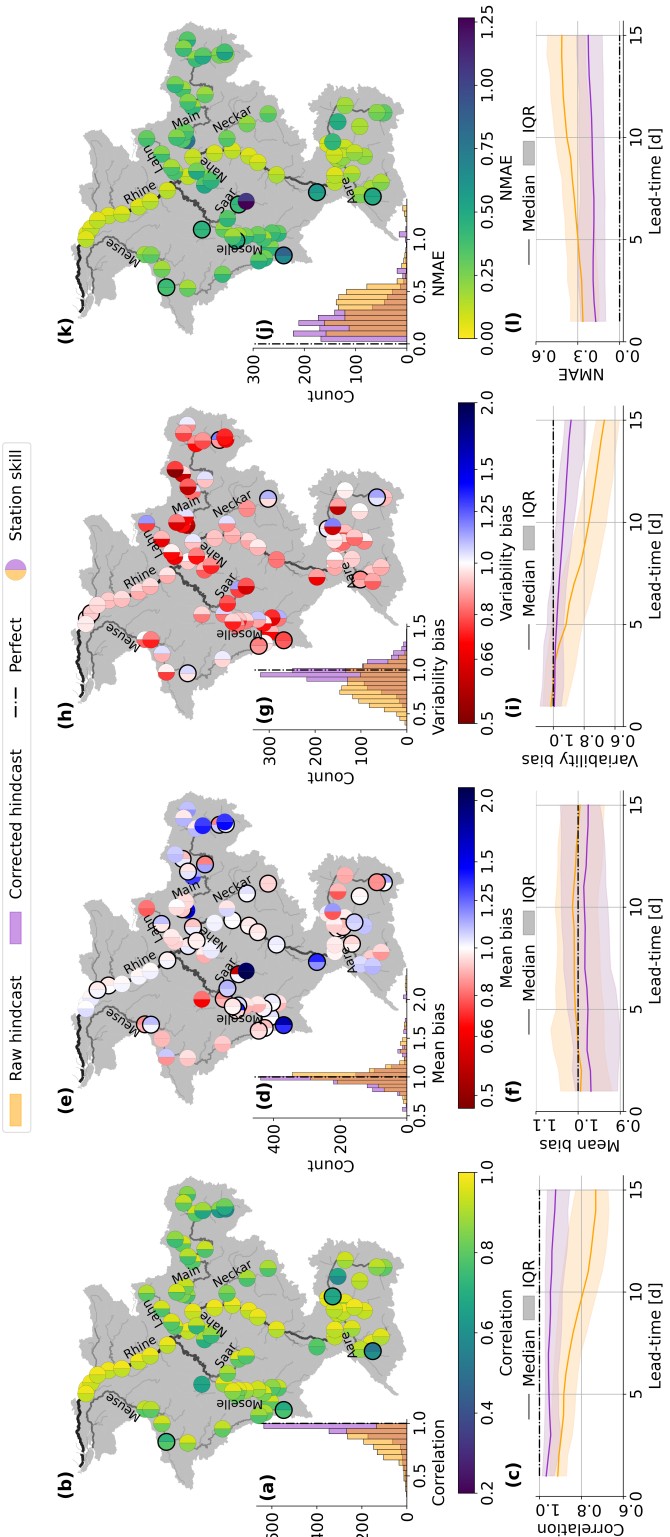

**Figure 6.** Skill of the ensemble means in terms of the correlation (a, b, c), mean bias (d, e, f), variability bias (g, h, i) and normalised mean absolute error (NMAE; j, k, l). The histograms show the distributions of the metrics pooled over all stations and lead-times for the raw (orange) and error-corrected (purple) ensembles (a, d, g, j). A perfect score for the metric is shown by the dashed black line. Catchment maps show the metric averaged across all lead-times at all 90 stations (b, e, h, k). The left (right) half of the marker shows the skill for the raw (error-corrected) ensemble. Black outlines indicate stations for which the updated ensemble has worse skill than the raw hindcast ensemble. Line plots show the distribution of the metric pooled over all 90 stations for each lead-time for the raw (orange) and error-corrected (purple) ensembles (c, f, i, l). The solid line shows the median value of the metric and the shaded region shows the interquartile range (IQR) of the metric. A perfect score for the metric is shown by the dashed black line.





clear spatial pattern emerges, as most rivers have a mix of improved and worsened stations. This spatial heterogeneity is also seen in the raw hindcast ensemble, with stations on the same river stretch often showing different biases. For example, stations

on the Neckar, and upstream of the Meuse show stations, that are under- and overestimated, as well as stations with very little bias. The heterogeneity suggests local factors which are not fully captured in the modelling system considerably influence flow bias. Stations showing the most improvement tend to have similar mean bias values to their neighboring stations in the raw hindcast ensemble, such as on the middle stretch of the Meuse, where four stations with similar biases show improvement (Fig. 6e). Spatial patterns of errors that are related to domain-wide model structure rather than local factors, such as weirs, are more

likely to be portrayed by the ensemble covariances allowing observational information to be more helpfully spread along the river network.

    The raw hindcast ensemble mean generally underestimates flow variability, with a variability bias below 1 (Fig. 6g). The error-corrected ensemble improves this, increasing the mean variability bias from 0.82 to 0.95, although the frequency of overestimation of flow variability is also increased (Fig. 6g). Stations where the error-corrected ensemble overestimates the

variability are often the most upstream station on their rivers (e.g., Plochigen station on the Neckar) or are much closer to downstream than upstream neighbours (e.g., Chooz station on the Meuse). This suggests the hindcast covariances between downstream stations and upstream locations are too large, causing excessive adjustment at upstream locations. Ten stations show worsened variability bias, including two stations downstream on the Rhine (Fig. 6h). The cause of the worsening of these two stations is the adjustment for the falling limb of a flood peak in July (Fig. 5b). Here, the hindcast uncertainty was very

small at short lead-times, causing the analysis to ignore observations and the error ensemble to remain relatively unchanged, despite changes in the error behavior following the peak.

    Overall, the error-corrected ensemble reduces the absolute error, with the average NMAE decreasing from 0.33 to 0.23 (Fig. 6j). The 7 stations with worsened NMAE are typically (5/7) the most upstream on their rivers (Fig. 6k; see discussion about correlation). Interestingly, the NMAE does not follow the same spatial pattern as the mean bias. The decrease in absolute

errors, despite an increase in mean bias, suggests that the error-corrected ensembles consistently underestimate flow, while the raw hindcast ensemble fluctuates more between under- and overestimation, which can compensate for each other in the mean bias metric.

    The raw and error-corrected ensemble means both decrease in skill in terms of correlation, variability bias, and NMAE with increasing lead-times. The raw hindcast ensemble loses skill more quickly in particular for lead-times longer than 5 days (Figs.

6c, 6i, and 6l). The uncertainty in the observations is not lead-time dependent. However, Fig. 3d shows that the ensemble covariances do change across lead-times, increasing for longer lead-times. The reduction in skill as lead-times increase suggest that the ensemble covariances are not able to spread the observational information along the river network as accurately at longer lead-times. This is likely due to an over estimation of the variance at longer lead-times.

### 7.2.2   Skill of the ensemble distribution

The reliability of the ensemble distribution is assessed using rank histograms at different lead times (Figs. 7a, 7b, and 7c). At short lead times, the raw hindcast ensemble is underdispersed, likely due to the use of a single set of initial conditions (Fig.





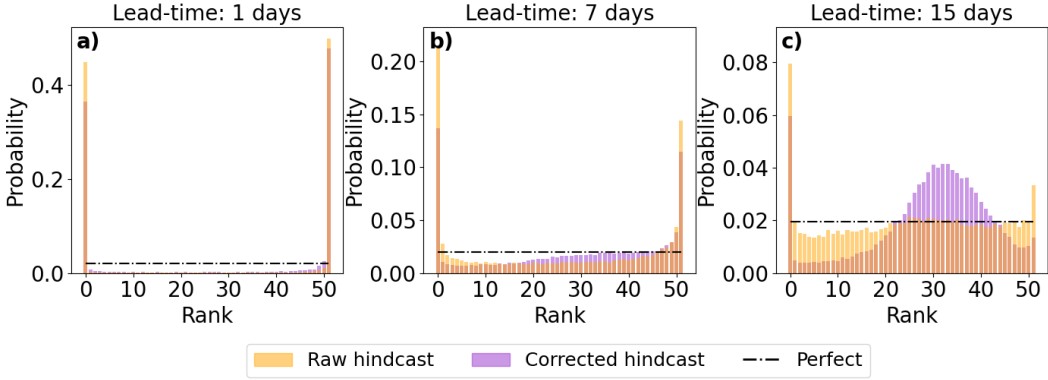

**Figure 7.** Reliability of the ensemble. Histograms show the rank of the ensemble pooled over all forecasts and stations for lead-times of 1 day (a), 7 days (b), and 15 days (c).

7a). Although the error-corrected ensemble shows slight improvement, it remains overconfident with minimal correction to the spread. Both the raw and error-corrected ensembles generally appear unbiased, with observations falling both above and below the ensemble predictions at similar frequencies. However, some bias may be masked by the narrow ensemble spread and it is

known that some stations are biased (Fig. 6b), likely contributing to the peaks at ranks 0 and 51 in the rank histograms.

As the lead-time increases, the spread of both ensembles becomes more reliable, and fewer observations fall outside the ensemble (Fig. 7b). However, even at a 15-day lead time, both ensembles show a tendency to overestimate observations, leading to a peak at rank 0, mostly due to a few stations consistently overestimating flow (Fig. 6b). Up to 7-day lead times, the rank histograms for both raw and error-corrected ensembles show similar shapes. Beyond 7 days, the raw hindcast ensemble's

histogram flattens, suggesting a reliable ensemble, while the error-corrected ensemble shows a peak around ranks 25-35, suggesting overdispersion (Fig. 7c). The left-skewness of the histograms is likely due to the inherent skewness in river discharge distributions. The LETKF update step seeks to minimise the difference between the ensemble mean and the true state of the system. The ensemble mean is often larger than the ensemble median leading to the observations falling in ranks above 25 if the adjustment method is successful and minimising the error of the mean (Figs. 5a and 5c).

As discussed in Section 4.1, the Kalman filter is not restricted to ensure positive discharge and there is therefore a need to adjusted the error ensemble before correction of the hindcast. Enforcing non-negative discharge was necessary, for example, for the Mainleus station on the Main for the hindcast generated on the 22 March 2021 (Fig. 5c). Whilst the ensemble mean is error-corrected at most lead-times, several members indicate river discharge values of 0 $m^3s^{-1}$. The river discharge is below 10 $m^3s^{-1}$ but a zero flow is unlikely in reality. This suggests the ensemble spread is not sufficiently corrected even though the

ensemble mean is improved as is also suggested by Fig. 7c.





## 8   Discussion

In general, the proposed data-assimilation-inspired method successfully spreads observational information along the river network improving the skill of the ensemble mean at ungauged locations. Locations downstream from assimilated observations are improved most although locations upstream are usually improved as well, even if they are far from neighbouring stations. This is likely due to two reasons: 1) constant biases in the river discharge estimates that are propagated downstream and hence can be accounted for when a downstream observation is assimilated, and 2) the daily aggregation of the river discharge extending the time period for which a downstream observations provides relevant information. If the error patterns of the ensemble mean at a location differ from those at nearby stations the method struggles to spread the observational information correctly. At shorter lead-times the reliability of the ensemble is slightly improved due to the decrease in the error of the ensemble mean. However, at longer lead-times the ensemble spread is often too large leading to an under-confident forecast.

Despite the method's ability to correct upstream, it could be beneficial to assimilate observations from as far upstream as possible. These observations do not necessarily need to be traditional in-situ observations but could come from Earth Observation (EO; Durand et al., 2023), crowdsourced or community observations (Le Coz et al., 2016; Etter et al., 2020), or camera based sensors (Vandaele et al., 2021). The key requirement is that an observation operator can be defined. Observation operators map the state of the system from state space to observations space. In our study the observation operator selects the grid-point that represents the location of the station on the modelled river network. The mapping of the station locations from the physical river network to the modelled river network is not trivial and several studies have attempted to automate this step (Isikdogan et al., 2017; Li et al., 2018). If this mapping is incorrect then representation errors can be introduced (Janjić et al., 2018). For example, if a station on a bypass channel is incorrectly located on the main channel. Observations from the station will undoubtedly provide erroneous information in the update step.

The covariance inflation method used here maintains consistency between the spread of the error ensemble and the spread of the hindcast (Section 5.2). This successfully stops the error ensemble from collapsing such that the observations are not ignored. However, in situations where the uncertainty of the hindcast ensemble is over- or under-estimated the covariance inflation does not correct the error ensemble covariances correctly. This can lead to the observations being ignored as for short lead-times in Fig. 5b, and could also be the cause for the slight degradation in skill of the ensemble mean in Fig. 6c, 6i, and 6l. Correcting the spread of the hindcast before using it in the inflation of the error covariances could solve this issue (Section 5.2). Covariance inflation techniques that use the innovation statistics could be used to first adjust the hindcast ensemble (e.g., Kotsuki et al., 2017). Alternatively, a lower threshold for the variance of the ensemble could be set - say 10% of the ensemble mean similarly to the observation error covariance matrix or the root mean square-error of the initial conditions. However, caution is needed not to artificially inflate the covariances too much such that the analysis increments become too large, in particular at short lead-times when the correlation is small (Fig. 3).

As discussed in Section 7.2.2, the resulting ensemble must be adjusted in some cases to avoid negative discharge values (Section 4.1). This does in some cases lead to ensemble members close to 0 $m^3 s^{-1}$ when a zero flow value is unlikely (Fig. 5c). This occurs due to the analysis increment being larger in magnitude than the value of some of the raw ensemble members.





In general, this is due to the skewed distribution of discharge (Bogner et al., 2012). Future work could look into applying anamorphosis to make the ensemble distribution more Gaussian-like (Nguyen et al., 2023). The covariances between grid boxes on larger rivers and station locations tend to be large even when the correlation is small. This is due to larger rivers having larger variances which is partially due to their larger river discharge magnitudes. Localisation does enforce a distance dependence on the covariance magnitudes but transforming the river discharge values to be comparable across the domain

may also help minimise the impact of overestimated ensemble spread. A transformation between river discharge and specific discharge (river discharge divided by upstream area) could be used to ensure that the ensemble covariances more accurately represent the true relationship between locations.

  The code developed for this study is designed to allow for research flexibility rather than operational efficiency. However, the error-adjustment of a single forecast took on average 8.5 minutes for the whole of the Rhine-Meuse catchment - a large

catchment. This suggests that, with proper parallelization, the method could be operationalized and applied to all gauged catchments in Europe. Before that, though, the method needs to be evaluated on additional catchments. The Rhine was selected because it is highly gauged, but this also means that the raw ensemble's skill is relatively high due to the effectiveness of the hydrological model calibration process. This could influence the method's performance in two ways: 1) the error ensemble may evolve more linearly than in less calibrated catchments, and 2) the hindcast ensemble's covariance may better represent

the covariances between the estimated errors. The next step should be applying this method to a catchment with lower skill than the Rhine.

  The method presented in this study spreads observation information along the river network but cannot yet be used as a post-processing method because observations from the hindcast period (the future) are assimilated. We envisage the method being developed further to make it applicable operationally as a hydrological forecast post-processing method. Nevertheless,

it may still be useful in certain situations, such as post-event analysis. After a flood event an assessment is often performed estimating the severity of the event as well as potential causes and mitigating factors. However, in-situ river gauges only present a snapshot of the event at specific locations and are often damaged during flood events, resulting in missing or incorrect data. EO estimations of river discharge could fill in some of the gaps but this would depend on the satellite's orbit and its availability at the right time (Douben, 2006). Reanalysis is another option, but it requires additional hydrological model runs

and may contain errors due to the structure of the hydrological model or errors in the meteorological observations. The method proposed here could offer a domain-wide estimate of observations without requiring additional model runs or a "warm-up" period typically needed in hydrological simulations to stabilize antecedent water storage within the catchment.

## 9 Conclusion

We present and evaluate a data-assimilation-inspired method for spreading observation information from gauged to ungauged

locations in a post-processing environment. This method enables the error-correction of an ensemble simulation at all grid boxes. The method utilises state augmentation within an LETKF framework to estimate an ensemble of error vectors. The error vectors are then used to correct each hindcast ensemble member separately.




Overall, the method successfully reduces the errors of the ensemble mean at ungauged locations in leave-one-out experiments. The adjusted ensemble mean has a higher correlation with the observed river discharge and is more able to capture the variability of the river discharge at a point. Whilst the magnitude of the errors are reduced the ensemble spread is not adjusted sufficiently resulting in an under-confident ensemble spread at longer lead-times. The adjusted ensembles are spatially and temporally consistent with the river discharge predictions showing smooth evolution both between grid-boxes on the same river and between lead-times. The method is most limited in its applications to locations further upstream than the assimilated observations and for hindcasts where the variance of the ensemble is incorrectly small which most often happens at shorter lead-times. These limitations can be minimised by further investigation into the localisation approach, for example having a different localisation length upstream and downstream from the assimilated observation, and the covariance inflation approach, which may involve applying a spread-correction to the hindcast ensemble as well as the error-ensemble.

Our method of spreading observation information could be used to improve post-event analysis. However, as the computational requirements and processing time are both small the method could also be developed further to allow for its application to the post-processing of operational forecasts. The prediction of river discharge at ungauged locations is a crucial challenge for hydrological research and once successfully achieved will allow for better preparedness for floods.

*Code and data availability.* The code used in this study for the error-correction method, evaluation of river discharge forecasts, and generation of the figures presented in this manuscript is available upon request. The river discharge forecast used in this study are from the Copernicus Emergency Management Service (CEMS)'s European Flood Awareness System (EFAS) and are available to download from https://ewds.climate.copernicus.eu/datasets/efas-forecast. The local drainage direction and channel length data is available from https://data.jrc.ec.europa.eu/dataset/f572c443-7466-4adf-87aa-c0847a169f23.

*Author contributions.* GM, HC, SD, and CP developed the method and designed the study. GM developed the software, performed the forecast error-correction and the evaluation, and drafted the manuscript. All the co-authors contributed to the editing of the manuscript, and to the discussion and interpretation of the results.

*Competing interests.* The authors declare that they have no competing interests.

*Acknowledgements.* We gratefully acknowledge the financial support provided by the following: Gwyneth Matthews is supported in part by the EPSRC DTP 2018-19 University of Reading (grant number: EP/R513301/1) and ECMWF. Hannah L Cloke is supported by the UKRI Natural Environment Research Council, Evolution of Global Flood Risk (EVOFLOOD; grant number: NE/S015590/1). Sarah L Dance is supported in part by the UKRI Natural Environment Research Council National Centre for Earth Observation (grant numbers: NE/X019063/1, NE/W004984/1, NE/Y006216/1).



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
