# Peer review of "Error-correction across gauged and ungauged locations: A data assimilation-inspired approach to post-processing river discharge forecasts"

_Hydrology and Earth System Sciences, 2024_

## Author Comment (AC1)

**Response to RC1 for hess-2024-3989: Matthews, G., et al. Error-correction across gauged and ungauged locations: A data assimilation-inspired approach to post-processing river discharge forecasts**

We thank the reviewer for their comments and suggestions which we believe will greatly improve the manuscript and strengthen the motivation for the method. The reviewer's comments have been summarised and numbered for clarity. The authors' responses are in blue.

1. The paper is difficult to follow due to dense mathematical notation and long, complex sentences.
   We will revisit the notation in the paper. We note that, where possible, we already use standard mathematical notation for data assimilation following Ide et al. (1997). We will add this reference to Section 2. We will reduce the number of symbols where possible, such as the symbol for the augmented observation operator and some superscripts. We will also remove the additional notation used to describe the approach for dealing with non-negative values and estimating the initial error ensemble as this notation is only used in Sections 4.1 and 5.3, and instead describe this with words.

   We will review the text in the manuscript and shorten particularly complex sentences throughout.

2. There is an overuse of jargon without sufficient introductory explanation for a broader hydrology audience.
   We will review the manuscript and remove unnecessary jargon, such as "spatiotemporal consistency". Where data assimilation technical terms are deemed by us to aid the description of the method, such as state augmentation, we will add a clear description of the term (see also comment 4).

3. The structure could be more concise, with a clearer division between methodology and results.
   Thank you for this comment. We will restructure Section 7 to make a clearer division between the results that show how the method works and the results that show the skill of the resulting ensemble. Where possible we will remove repetition and unnecessary detail (see comment 12).

4. The state augmentation approach is described in a way that makes the approach seem unnecessarily complex.
   We will add the following sentences: "*State augmentation is a technique used in data assimilation to estimate the state and parameters of a system simultaneously. An augmented state is defined by appending the parameters to the state vector allowing both to be updated by the data assimilation method.*" We will also move the definition of the error ensemble to Section 2, streamlining the state augmentation description.

5. The assumption of constant error propagation is not well justified.
   We will add the following to Section 3.1 and add a discussion of this assumption to Section 7: "As the true evolution of the error vectors at all grid-boxes is unknown, we assume a simple persistence model, such that $\mathbf{b}_k^{(i)} = \mathbf{b}_{k-1}^{(i)}$. This is a common assumption used in state augmentation (Pauwels et al., 2020; Rasmussen et al., 2016; Ridler et al., 2018; Martin, 2001)."

6. Also, related to this, the use of precomputed model outputs instead of an evolving state might introduce additional errors, which are not sufficiently discussed.
We will amend the discussion of this assumption (line 219) to: "The assumptions made in Eqs. (18) and (19) make our system sub-optimal from a data assimilation perspective but are necessary to avoid rerunning the hydrological model. Importantly, we aim to estimate the error of the precomputed model output at each lead time. Therefore, while the lack of state evolution makes the hindcast component update sub-optimal, the update of the error ensemble remains mathematically consistent".

7. Inflation:
We will restructure Section 5.2 to address the following comments.
   i    I find the inflation method to be heuristic with little to no mathematical rigor. For instance, the assumption that the hindcast variance is a proxy for error growth does not account for potential biases in the raw ensemble itself.
   The reviewer is correct that the covariance inflation technique used is a heuristic method. We initially tried a simpler multiplicative inflation but found that this was not suitable due to the large variations in hindcast spread as a function of lead-time.
   Our new approach is a practical solution to the issue of filter divergence that is inspired by blending techniques such as the RTPP method (Zhang et al., 2004). We will add a comment to Section 5.2 to make this clearer. The limitations of using the hindcast uncertainty as a proxy for the uncertainty in the error estimate are discussed in the discussion section (lines 646-656).

   ii   Unlike RTPP, the proposed inflation blends analysis and "estimated" perturbation information without explicitly evolving them. What motivates such an approach?
   Our goal is to use this approach for post-processing with pre-computed ensemble forecasts. Explicit online evolution of perturbations is not feasible in this scenario. Hence, the error ensembles are evolved between timesteps using a persistence model (see comment 5). Blending the evolved perturbations with an estimated perturbation method is inspired by palaeoclimatological reanalysis work such as Valler et al., (2019) where a climatological error-covariance matrix is blended with the background error-covariance matrix. Rather than a climatological matrix we use "estimated" perturbations. During the development of this method, it was found that the "estimated" perturbation matrix must 1) preserve the spatial structure of the river network and 2) must be adaptive, to avoid filter divergence and maintain physically plausible estimates along the river network. The hindcast ensemble perturbation matrix satisfies both these requirements. We will add this motivation to Section 5.2.

   iii  The inflation parameter, alpha, is computed from a 3 steps-average of the hindcast (eq. 28). Why this choice is appropriate? I recommend testing with different alpha values through sensitivity experiments.
   Sensitivity experiments were conducted in the development of this method, but we did not include the results in the original manuscript for brevity. Our analysis indicated that 1) a constant alpha value was not suitable at different lead-times due to the change in hindcast spread, and 2) a lead-time dependent constant alpha value was not suitable for different flow situations. We therefore selected a method that is forecast dependent. An average across 3 steps was selected to ensure a smoothly changing alpha mitigating instabilities. We will add comments along these lines to Section 5.2.

iv    If inflation is not localized along the network, that should be clarified and justified.

The inflation factor does not vary in space (although it does vary in time). It is applied to perturbation matrices that themselves vary in space, consistent with the river network. Thus, the inflated covariances provide physically plausible error-variances and error-correlations between locations. While spatially varying inflation methods are available, (e.g., Kotsuki et al, 2017) this would result in many more inflation parameters to estimate, and the results may not be spatially consistent as each element gets inflated separately. We will make this clear in Section 7 (discussion).

8.  Spread: It's clear that the method tends to overcorrect at short lead times but yields underconfident ensembles at longer lead times (as shown in Figs. 4, 5). In general, one expects the ensemble spread to accurately represent the forecast uncertainty but the issues the authors face could be related to the ad-hoc inflation.

We agree with the reviewer that the inflation method is a reason for limited reliability of the ensemble-spread at longer lead-times. However, we also note that it is very rare that the ensemble spread accurately represents the forecast uncertainty for all lead times and locations (e.g., see Kotsuki et al, (2017)'s comparison of inflation approaches). For our proof-of-concept study, we have not carried out extensive tuning experiments, instead making some pragmatic choices. A study comparing different inflation approaches for the context of post-processing ungauged locations is left for future work. The limitations of using the inflation method are discussed on lines 646-656. We will extend the discussion on the spread in Section 7.

9.  I would also note that real hydrological errors are dynamic, but the paper assumes the errors to remain constant between cycles. A flow-dependent error propagation model and perhaps an adaptive inflation approach could address these issues.

The error covariance propagation is flow dependent (based on an ensemble of precomputed hind-casts) and the inflation factor is adaptive in time. Please see our response to comments 5 and 7.iii.

10. Localization: The choice of the length scale (262 km) should be better justified. There is no sensitivity analysis to determine whether this choice is optimal or whether smaller/broader radius would improve the results.

Sensitivity experiments were conducted during the development of this method. It was found that the optimal length scale varied by location, lead-time, and tuning metric of choice, but overall the differences were small for length scales from 65km to 786km. We therefore decided to define the length scale as the maximum distance between any grid-box and its closest river gauge which for our case study is 262km. This definition ensures all grid-boxes are updated by the LETKF and allows the method to be transferred to other catchments and models without the need to perform computationally expensive tuning experiments. We will make this clear in Section 5.1.

11. Also tangential to this, the authors need to revisit the equal error correction assumption in upstream and downstream locations. Overall, upstream locations are less dependent on distant downstream observations. Obviously, downstream conditions are often affected by accumulating upstream flows.

The propagation of the error-correction along the river network is determined by the ensemble covariances and the localisation applied. This means that while the localisation length is equal

upstream and downstream, the actual analysis increments are not. This can be seen in Figure 2 where we show the analysis increments for single observation experiments. We will add comments to the discussion of Figure 2 to make this point clearly.

We agree with the reviewer that the relationships upstream and downstream are different. This is shown in Figure 3a. The cross-correlations between the hindcast and error ensembles are strongest along the river stretch near the station and decrease at longer distances. The larger correlations downstream of the station are along the flow path of the river whereas upstream the correlations show a more branch like structure because the station location is impacted by accumulation of flows from all upstream tributaries. We will add comments to the discussion of Figure 3 to make this point clearly.

The localisation also dampens the influence of distant observations. Emery et al., (2020) investigate the use of localisation along the river network. They found that an observation can be beneficial to both upstream and downstream locations particularly for distances for which the flow transit time is less than the time between analyses. We will extend the discussion regarding the localisation in Section 7.

12. Figures: The figures are well-intended but too dense and overloaded with information, making them difficult to interpret and extract keys findings. I suggest splitting the complex ones (e.g., Figs. 3, 4, 6) and definitely simplify the annotations
Thank you for this comment. To address this comment, and comment 1 of RC2, we will reduce the content of some of the figures. We will make the following changes to the figures:
- Figure 3: We will remake this figure removing panels c and g. Some of the information from c and g is duplicated in the hydrographs in Figure 5. Removing these panels will shorten the discussion (see comment 1 from RC2). This will also allow more room for remaining panels of Figure 3 making key details clearer.
- Figure 4: We will simplify the annotations as request by the reviewer. We will combine the legends making the comparison between panels easier and more clearly indicate the difference between rows 1 and 2.
- Figure 6: We will remove the panels a, d, g, and j, and related discussion as this information is also shown in the remaining panels. We will also remove the river names, which are already shown in Figure 3. We will combine the legends of panels c, f, i, and l and clearly label the metric shown in each column.

13. Line 6: "Error vector for each ensemble members" seems vague and unclear.
Thank you for this comment. We will change this sentence to read: "Our new method employs state augmentation within the framework of the Local Ensemble Transform Kalman Filter (LETKF). Using the LETKF, an error vector representing the forecast residual is estimated for each ensemble member."

14. Line 12: The term "proxy" could mean a lot of different things. Clarify the nature of updates, whether that's real data assimilation experiment or an OSSE.
Our experiment uses real gauge-data and meteorological forcings and we will make this clear in the introduction. We will change this statement to read "A spatial cross-validation strategy is used to assess the ability of the method to spread the correction along the river network to ungauged locations"

15. Line 160: I would use "cycled" instead of "iterated"

   Thank you. This will be changed.

16. Line 160: Replace "at each timestep" with "at each observation time"

   We use "timestep" rather than "observation time" as the analysis times are dictated by the availability of the precomputed hindcast data as well as by the availability of observations. For clarity, we will change it to "at each hindcast timestep for which observations are available".

17. Line 178: Replace "weights" with "weighs"

   Thank you. We will rephrase this sentence.

18. There are too many "see section xxx". This made navigation frustrating; I kept going back and forth. Consider restructuring for better flow.

   We will remove some cross-references and reword others to e.g., "(described in Section X)" as a signpost for the reader for the more novel components of the method (see comment 8 in RC2).

19. The word "improved" is overused in my opinion. Consider other synonyms "enhanced", "refined", ...

   We will reword sentences where appropriate to be more specific about the effect being described.

20. Explain technical terms more clearly, for instance "spatiotemporal consistency"

   See comment 3.

References

Emery, C. M., David, C. H., Andreadis, K. M., Turmon, M. J., Reager, J. T., Hobbs, J. M., ... & Rodell, M. (2020). Underlying fundamentals of Kalman filtering for river network modeling. *Journal of Hydrometeorology*, *21*(3), 453-474. https://doi.org/10.1175/JHM-D-19-0084.1

Ide, K., Courtier, P., Ghil, M., & Lorenc, A. C. (1997). Unified notation for data assimilation: Operational, sequential and variational (gtspecial issueltdata assimilation in meteology and oceanography: Theory and practice). *Journal of the Meteorological Society of Japan. Ser. II*, *75*(1B), 181-189. https://doi.org/10.2151/jmsj1965.75.1B_181

Kotsuki, S., Ota, Y., & Miyoshi, T. (2017). Adaptive covariance relaxation methods for ensemble data assimilation: Experiments in the real atmosphere. *Quarterly Journal of the Royal Meteorological Society*, *143*(705), 2001-2015. https://doi.org/10.1002/qj.3060

Martin, M. J. (2001). *Data assimilation in ocean circulation models with systematic errors* (Doctoral dissertation, University of Reading).

Pauwels, V. R., Hendricks Franssen, H. J., & De Lannoy, G. J. (2020). Evaluation of State and Bias Estimates for Assimilation of SMOS Retrievals Into Conceptual Rainfall-Runoff Models. *Frontiers in Water*, *2*, 4. https://doi.org/10.3389/frwa.2020.00004

Rasmussen, J., Madsen, H., Jensen, K. H., & Refsgaard, J. C. (2016). Data assimilation in integrated hydrological modelling in the presence of observation bias. *Hydrology and earth system sciences*, *20*(5), 2103-2118. https://doi.org/10.5194/hess-20-2103-2016

Reichle, R. H., D. B. McLaughlin, and D. Entekhabi, 2002: Hydrologic Data Assimilation with the Ensemble Kalman Filter. *Mon. Wea. Rev.*, 130, 103–114, https://doi.org/10.1175/1520-0493(2002)130<0103:HDAWTE>2.0.CO;2.

Ridler, M. E., Zhang, D., Madsen, H., Kidmose, J., Refsgaard, J. C., & Jensen, K. H. (2018). Bias-aware data assimilation in integrated hydrological modelling. *Hydrology Research*, *49*(4), 989-1004. https://doi.org/10.2166/nh.2017.117

Valler, V., Franke, J., & Brönnimann, S. (2019). Impact of different estimations of the background-error covariance matrix on climate reconstructions based on data assimilation. *Climate of the Past*, *15*(4), 1427-1441. https://doi.org/10.5194/cp-15-1427-2019

Zhang, F., Snyder, C., & Sun, J. (2004). Impacts of initial estimate and observation availability on convective-scale data assimilation with an ensemble Kalman filter. *Monthly Weather Review*, *132*(5), 1238-1253. https://doi.org/10.1175/1520-0493(2004)132<1238:IOIEAO>2.0.CO;2

---

## Author Comment (AC2)

**Response to RC2 for hess-2024-3989: Matthews, G., et al. Error-correction across gauged and ungauged locations: A data assimilation-inspired approach to post-processing river discharge forecasts**

We thank the reviewer for their insightful comments and helpful suggestions which we believe will greatly strengthen the evaluation and discussion of the new method. The reviewer's comments have been summarised and numbered for clarity. The authors' responses are in blue.

1. The paper is too long, in particular the description of the methods.
   We will shorten the descriptions of methods by restructuring and condensing the descriptions in Sections 4.1, and 5.1-5.3. Addressing some of the comments from both reviewers such as comment 1, 3, and 12 from RC1 will also reduce the length of the manuscript.

2. I am a bit concerned about how applicable these methods are outside the case study attempted (see specific comments below). For example, the use of a very large catchment is likely to allow the authors to make simplifying assumptions such as that residuals will be normally distributed, or that errors can be characterised using a 10-day window. I would be interested in some discussion of how generalisable these methods are.
   Thank you for this comment. We will extend the discussion on the generalisability of the method to include the estimation of the initial error ensemble (where the 10-day window is used) and the assumption of Gaussianity. See comments 5, 6, and 18.

3. L58 "ensemble Kalman Filters are common data assimilation methods for hydrological applications" this is true for hydrological research, but (to me at least) it remains a curiosity as to why data assimilation within hydrological models - including with ensemble Kalman Filters - remains to my knowledge quite rare in operational streamflow forecasting systems.
   This is a good point. We will specify in "hydrological research applications". In our experience, data assimilation cannot be used in large-scale operational hydrological forecasting due to data latency issues. For weather forecasting, significant international infrastructure has been set up to share and distribute observations rapidly via the GTS, so that observations can be processed in the critical path for forecast production. For gauge data, some observations are available in near-real time, but many are distributed much more slowly. In this study we investigate the use of data assimilation techniques within a post-processing environment as a first step towards an operationalizable post-processing method for ungauged locations. We will make this clearer in the introduction.

4. L90 "Hydrological ensemble forecasts consist of N potential realizations referred to as ensemble members" I think it would be good to state explicitly which variable(s) you are discussing here, as it wasn't clear to me - I'm assuming streamflow (or runoff, as it's on a grid?)?
   The variable of interest is streamflow or river discharge. We will change this to "The hydrological ensemble forecasts consist of N potential realizations of future river discharge referred to as ensemble members."

5. L107 I would have thought with a strongly skewed (and potentially zero bounded) variable like streamflow, an additive error only generally holds after a normalising transformation has been applied (and, if applicable, zero values have been dealt with).
   We will add normalising transformations to the discussion (see comment 18). The reviewer is correct that the assumption of Gaussianity limits the applicability of an additive error by resulting occasionally resulting in negative discharges. We deal with negative discharge values

as described in Section 4.1. The impact and potential solutions are discussed in Sections 6.2 and 7.

6. L112 Similar to the above criticism at L107, Equation 6 appears to assume that errors are normal and homoscedastic. If my understanding of what is being assimilated is correct, this is highly unlikely to hold for streamflow, for which residuals are almost always non-normal and heteroscedastic. See e.g. Smith et al. 2015, among many others.
We agree that streamflow residuals are often non-Gaussian and heteroscedastic. Our framework does employ updates based on Gaussian assumptions as we use the LETKF, but the resulting distribution is not necessarily Gaussian (Reichle et al., 2002). However, we do not assume homoscedasticity: the ensemble spread evolves dynamically and reflects state-dependent and lead-time-dependent error variability. We will change Equation 6 to express the distribution of the updated ensemble more accurately.

7. L145 "we adopt the common assumption that the error is constant" I would not have said this is common. I would say it's much more common to use autoregressive models (often AR1) to describe the autocorrelation between residuals in streamflow. I understand why this is a pragmatic simplification, but errors often do change with lead time as the value of forecast information decays.
Thank you for the comment. We will add the following to Section 3.1 and add a discussion of this assumption (and possible developments) to Section 7: "As the true evolution of the error vectors at all grid-boxes is unknown, we assume a simple persistence model, such that $\mathbf{b}_k^{(i)} = \mathbf{b}_{k-1}^{(i)}$. This is a common assumption used in state augmentation (Pauwels et al., 2020; Rasmussen et al., 2016; Ridler et al., 2018; Martin, 2001)."

8. L149 "define the propagation" I'm not sure what 'propagation' means here, given the error is assumed constant in time. Can the authors clarify? Nevermind - the authors do this in Section 3.2! The authors may want to flag that the explanation for this is coming.
We will highlight that the propagation equation defined on line 150 if for use in the LETKF described in Section 3.2

9. L180 "(see Eqs. (8) and (9) in Bell et al., 2004)" I feel that if the authors need to specify equations from another study to describe these methods, the equations should be present in the paper (in an appendix is fine) - especially Eq (9) of Bell et al. which the authors later describe as 'key' to the method. (Unless they are included later?)
We will add the decomposition of the Kalman gain matrix as an appendix.

10. Figure 1 - this is a really nice, clarifying figure.
Thank you!

11. L223 "We enforce non-negativity by further adjusting the error ensemble members after the LETKF update step (Fig 1)." This indicates that zero values are present in output state, indicating that errors are not continuously distributed. I realise not everyone handles zeros, but it would be good to acknowledge the limitation of this assumption (as noted above).
We will extend the discussion regarding the assumption of Gaussianity for river-discharge. See comment 5.

12. L265 "Eq. 4.10 in Gaspari and Cohn (1999)" - I think the authors should include this equation, as well as discussing (briefly) why they thought the form of this equation appropriate for this task. The regionalisation of errors is in my view the major contribution of the paper.

The Gaspari and Cohn correlation function is a commonly used localisation function in data assimilation as it smoothly decreases to a definable radius. We will add the Gaspari and Cohn function as an appendix.

13. L272 "We propose instead for the localisation length scale to be defined as the maximum distance between any grid point and its closest observation." This seems like a sensible choice.

Thank you.

14. L325 "(here 10 days)" This is a long period over which to assess an error - some use periods of this length for bias correction (e.g. Bennett et al. 2021). I'm assuming this really only works for larger catchments where rivers have slower varying errors; I would have thought for small headwater gauges shorter periods would be more appropriate. It also explains why errors are assumed not to vary with lead time, above. This is all fine, but the authors may wish to mention this in their discussion.

This is a good point. The 10-day period is used to generate the initial error ensemble mean. This initial estimate is not used to correct the river discharge ensembles directly, but rather to provide a starting point for the LETKF. The LETKF then updates the error ensemble at each timestep. We will update Figure 1 to better clarify this process (see Figure below).

We selected a 10-day period to capture the consistent biases of the hydrological model but also to allow for seasonal/dynamic variation in this bias. We agree that shorter periods may be more appropriate for smaller, fast-responding catchments and will clarify this in the discussion.

[Figure]

15.

   a. L408 "we assume that the observation errors from different gauge stations are uncorrelated" I'm not suggesting a change here, and I think this is a reasonable suggestion without additional information. But I suspect the long-range nature of the errors (a 10 day period) may undermine the assumption somewhat.

   The observation errors we refer to on line 408 are the errors described in Equation 7. The sources of these errors are instrument uncertainty, observation processing, observation operator error and scale mismatch between the observations and the model resolution.

Observation errors are assumed to be uncorrelated with the prior errors which is a standard assumption in data assimilation. We will clarify this in the text.

b. I'm also curious what happens when errors are propagated in space: what happens when you get a point equidistant (or close to equidistant) from two gauges, and the errors from the two gauges interact in some way (e.g. cancel each other, or sum).
The Kalman gain matrix governs the spreading of observation information in space. A weighted mean is calculated. The weight of an observation is determined by the cross-covariances between the hindcast and error ensembles, the distance from the observation (via the localisation), and the uncertainty in the observation itself. The left and central panels in Figure 2 show single observation experiments, indicating how information from one observation is spread spatially. The panels on the right of Figure 2 show how observation information is propagated when all available observations are assimilated. A study into the impact of the specific locations of the observations is left for future work. We will add this discussion to Section 7.

16. L438 "forecast mean is decreased by the proposed method we use the Normalised Mean Absolute Error" It's preferable to apply measures of absolute error to the ensemble median. See, e.g., Taggart (2022).
Thank you for this very helpful comment. As it is the ensemble mean that we want to evaluate we will use the normalised RMSE and make the appropriate changes to Sections 6.4 and 7.1.

17. L526 "However, this assumption is necessary to propagate the hindcast to the next time step without the use of a hydrological model (Section 3.1)." Perhaps, but one application of ensemble predictions is to sum ensemble members through time (e.g. to assess cumulative inflows to reservoirs). From this figure, it seems this would result in highly unreliable accumulations. This may not be an application of EFAS (I don't know), but if the method is to have more general applicability this is a serious weakness.
The hydrographs shown in Figures 4b and 4e are not the final hydrographs resulting from this method but instead are an intermediate step used to investigate the impact of the methodological assumptions made. We will restructure Section 7 to make a clearer division between the results that show how the method works and the results that show the skill of the resulting ensemble (see comment 3 from RC1).

18. L660 "Future work could look into applying anamorphosis to make the ensemble distribution more Gaussian-like" I'm not familiar with the concept of anamorphosis, but a conventional way of doing this is to use normalising transformations, of which many are available for hydrological variables.
Anamorphosis is very similar to normalising transforms used in hydrology. We will add the use of normalising transformations to this paragraph.

19-23. Typos and grammatical errors.
These typos will all be corrected. Many thanks to the reviewer for catching them!

References
Martin, M. J. (2001). *Data assimilation in ocean circulation models with systematic errors* (Doctoral dissertation, University of Reading).

Pauwels, V. R., Hendricks Franssen, H. J., & De Lannoy, G. J. (2020). Evaluation of State and Bias Estimates for Assimilation of SMOS Retrievals Into Conceptual Rainfall-Runoff Models. *Frontiers in Water*, *2*, 4. https://doi.org/10.3389/frwa.2020.00004

Rasmussen, J., Madsen, H., Jensen, K. H., & Refsgaard, J. C. (2016). Data assimilation in integrated hydrological modelling in the presence of observation bias. *Hydrology and earth system sciences*, *20*(5), 2103-2118. https://doi.org/10.5194/hess-20-2103-2016

Reichle, R. H., D. B. McLaughlin, and D. Entekhabi, 2002: Hydrologic Data Assimilation with the Ensemble Kalman Filter. *Mon. Wea. Rev.*, **130**, 103–114, https://doi.org/10.1175/1520-0493(2002)130<0103:HDAWTE>2.0.CO;2.

Ridler, M. E., Zhang, D., Madsen, H., Kidmose, J., Refsgaard, J. C., & Jensen, K. H. (2018). Bias-aware data assimilation in integrated hydrological modelling. *Hydrology Research*, *49*(4), 989-1004. https://doi.org/10.2166/nh.2017.117

---

## Author Response (AR1)

Response to RC1 for hess-2024-3989: Matthews, G., et al. Error-correction across gauged and ungauged locations: A data assimilation-inspired approach to post-processing river discharge forecasts

We thank the reviewer for their comments and suggestions which we believe will greatly improve the manuscript and strengthen the motivation for the method. The reviewer's comments have been summarised and numbered for clarity. The authors' responses are in blue. Line number, sections, and figures refer to revised manuscript.

1. The paper is difficult to follow due to dense mathematical notation and long, complex sentences.

We have revised some of the notation in the paper. We note that, where possible, we already used standard mathematical notation for data assimilation following Ide et al. (1997) and have added this reference to Section 2 (line 93). We have reduced the number of symbols where possible, such as the symbol for the augmented observation operator and some superscripts. We have also removed the additional notation used to describe the approach for dealing with non-negative values and estimating the initial error ensemble (and instead describe these processes with words) as this notation was only used in Sections 4.1 and 5.3, respectively.

We have shortened particularly complex sentences throughout.

2. There is an overuse of jargon without sufficient introductory explanation for a broader hydrology audience.

We have removed unnecessary jargon, such as "spatiotemporal consistency". Where data assimilation technical terms were deemed by us to aid the description of the method, such as state augmentation, we have added a clear description of the term (see also comment 4).

- 3. The structure could be more concise, with a clearer division between methodology and results. Thank you for this comment. We have restructured Section 7 to make a clearer division between the results that show how the method works (Section 7.1) and the results that show the skill of the resulting ensemble (Section 7.2). Where possible we have removed repetition and unnecessary detail (see comment 12).
- 4. The state augmentation approach is described in a way that makes the approach seem unnecessarily complex.

We have added the following sentences (lines 133-134): "State augmentation is a technique used for online bias-correction in data assimilation that allows the simultaneous estimation of the system state and biases. An augmented state is defined by appending the biases to the state vector, allowing both to be updated by the data assimilation method." We have also moved the definition of the error ensemble to Section 2, streamlining the state augmentation description.

5. The assumption of constant error propagation is not well justified. We have added the following to Section 3.1 (lines 146-148): "As the true evolution of the error vectors at all grid-boxes is unknown, we assume a simple persistence model, such that  $\mathbf{b}_k^{(i)} = \mathbf{b}_{k-1}^{(i)}$ . This is a common assumption used in state augmentation (Pauwels et al., 2020; Rasmussen et al., 2016; Ridler et al., 2018; Martin, 2001)." We have also added a discussion of

this assumption to Section 7 (lines 671-678).

6. Also, related to this, the use of precomputed model outputs instead of an evolving state might introduce additional errors, which are not sufficiently discussed.

We have extended the discussion of this assumption on lines 214-217: "The assumptions made in Eqs. (18) and (19) make our system sub-optimal from a data assimilation perspective but are necessary to avoid rerunning the hydrological model. Importantly, we aim to estimate the error of the precomputed model output at each lead time. Therefore, while the lack of state evolution makes the hindcast component update sub-optimal, the update of the error ensemble remains mathematically consistent".

**7. Inflation:**

We have restructured Section 5.2 to address the following comments.

i I find the inflation method to be heuristic with little to no mathematical rigor. For instance, the assumption that the hindcast variance is a proxy for error growth does not account for potential biases in the raw ensemble itself.

The reviewer is correct that the covariance inflation technique used is a heuristic method. We initially tried a simpler multiplicative inflation but found that this was not suitable due to the large variations in hindcast spread as a function of lead-time. Our new approach is a practical solution to the issue of filter divergence that is inspired by blending techniques such as the RTPP method (Zhang et al., 2004). We have added the following sentence to Section 5.2 to make this clearer (lines 278-281): "We implement a heuristic covariance inflation method inspired by the *relaxation-to-prior perturbations* technique (Zhang et al., 2004; Kotsuki et al., 2017). However, as we are working within a post-processing context, we adapt the method for use with predefined ensembles (i.e., without evolving the inflated perturbations between timesteps)."

The limitations of using the hindcast uncertainty as a proxy for the uncertainty in the error estimate are discussed in the discussion section (lines 636-647).

- ii Unlike RTPP, the proposed inflation blends analysis and "estimated" perturbation information without explicitly evolving them. What motivates such an approach? Our goal is to use this approach for post-processing with pre-computed ensemble hindcasts. Explicit online evolution of the inflated perturbations (as is required in the traditional RTPP method) is not feasible in this scenario. Blending the ensemble perturbation matrix with an estimated perturbation matrix is inspired by palaeoclimatological reanalysis work such as Valler et al., (2019), where a climatological error-covariance matrix is blended with the ensemble error-covariance matrix. We add the following justification to Section 5.2 for our choice of estimated matrices (lines 290-292): "During development, it was found that the estimated matrices must have spatial structures consistent with the river network and be forecast and lead-time-dependent. For simplicity, and as the raw hindcast perturbations satisfy these requirements, we set both  $\mathbf{X}_{k+1}^{est}$  and  $\mathbf{B}_{k+1}^{est}$  equal to the raw hindcast perturbation matrix (Dee, 2005; Martin et al., 2002)."
- iii The inflation parameter, alpha, is computed from a 3 steps-average of the hindcast (eq. 28). Why this choice is appropriate? I recommend testing with different alpha values through sensitivity experiments.
  - Sensitivity experiments were conducted in the development of this method, but we did not include the results in the original manuscript for brevity. Our analysis indicated that

- 1) a constant alpha value was not suitable at different lead-times due to the change in hindcast spread, and 2) a lead-time dependent constant alpha value was not suitable for different flow situations. We therefore selected a method that is forecast dependent. An average across 3 steps was selected to ensure a smoothly changing alpha mitigating instabilities. We have added this justification to Section 5.2 (lines 301-303).
- iv If inflation is not localized along the network, that should be clarified and justified. The inflation factor does not vary in space (although it does vary in time). We add the following sentence for clarity (lines 303-305): "While \$\alpha\$ is not spatially varying, it is applied to perturbation matrices with spatial structures consistent with the river network, ensuring physically plausible ensemble perturbations."
- 8. Spread: It's clear that the method tends to overcorrect at short lead times but yields underconfident ensembles at longer lead times (as shown in Figs. 4, 5). In general, one expects the ensemble spread to accurately represent the forecast uncertainty but the issues the authors face could be related to the ad-hoc inflation.
  We agree with the reviewer that the inflation method is a reason for limited reliability of the ensemble-spread at longer lead-times. However, we also note that it is very rare that the ensemble spread accurately represents the forecast uncertainty for all lead times and locations (e.g., see Kotsuki et al, (2017)'s comparison of inflation approaches). For our proof-of-concept
  - study, we have not carried out extensive tuning experiments, instead making some pragmatic choices. A study comparing different inflation approaches for the context of post-processing ungauged locations is left for future work. The limitations of using the inflation method are discussed on lines 636-647. We have extended the discussion on the spread in Section 7.1.2.
- 9. I would also note that real hydrological errors are dynamic, but the paper assumes the errors to remain constant between cycles. A flow-dependent error propagation model and perhaps an adaptive inflation approach could address these issues.
  The error covariance propagation is flow dependent (based on an ensemble of precomputed hindcasts) and the inflation factor is adaptive in time. Please see our response to comments 5 and 7.iii.
- 10. Localization: The choice of the length scale (262 km) should be better justified. There is no sensitivity analysis to determine whether this choice is optimal or whether smaller/broader radius would improve the results.
  - The length scale is defined as the maximum distance between any grid-box and its closest river gauge which for our case study is 262km. This definition ensures all grid-boxes are updated by the LETKF and allows the method to be transferred to other catchments and models without the need to perform computationally expensive tuning experiments. Sensitivity experiments conducted during the development of this method found that the optimal length scale varied by location, lead-time, and tuning metric of choice, but overall, the differences were small for length scales from 65km to 786km. This clarification is provided on lines 268-273.
- 11. Also tangential to this, the authors need to revisit the equal error correction assumption in upstream and downstream locations. Overall, upstream locations are less dependent on distant downstream observations. Obviously, downstream conditions are often affected by accumulating upstream flows.

The propagation of the error-correction along the river network is determined by the ensemble covariances and the localisation applied. This means that while the localisation length is equal upstream and downstream, the actual analysis increments are not. This can be seen in Figure 2 where we show the analysis increments for single observation experiments. We have added a discussion on the impact up- and downstream from the observation on lines 449-463.

We agree with the reviewer that the relationships upstream and downstream are different. This is shown in Figure 3a. The cross-correlations between the hindcast and error ensembles are strongest along the river stretch near the station and decrease at longer distances. The larger correlations downstream of the station are along the flow path of the river whereas upstream the correlations show a more branch like structure because the station location is impacted by accumulation of flows from all upstream tributaries. We have added comments to the discussion of Figure 3 to make this point clearly (lines 482-488)

The localisation also dampens the influence of distant observations. Emery et al., (2020) investigate the use of localisation along the river network. They found that an observation can be beneficial to both upstream and downstream locations particularly for distances for which the flow transit time is less than the time between analyses. We have extended the discussion regarding the localisation in Section 8 (lines 628-635).

12. Figures: The figures are well-intended but too dense and overloaded with information, making them difficult to interpret and extract keys findings. I suggest splitting the complex ones (e.g., Figs. 3, 4, 6) and definitely simplify the annotations

Thank you for this comment. To address this comment, and comment 1 of RC2, we have reduced the content of some of the figures. We have made the following changes to the figures:

- Figure 3: We have removed panels c and g from this figure. Some of the information from c and g was duplicated in the hydrographs in Figure 5. This has allowed more room for remaining panels of Figure 3 making key details clearer.
- Figure 4: We have simplified the annotations as request by the reviewer. We have combined the legends making the comparison between panels easier and have more clearly indicated the difference between rows 1 and 2.
- Figure 6: We have removed panels a, d, g, and j, and the related discussion as this information is shown in the remaining panels. We have also removed the river names, which are already shown in Figure 2. We have combined the legends of panels c, f, i, and l and clearly labelled the metric shown in each column.
- 13. Line 6: "Error vector for each ensemble members" seems vague and unclear.

Thank you for this comment. We have changed this sentence (lines 4-6): "Our new method employs state augmentation within the framework of the Local Ensemble Transform Kalman Filter (LETKF). Using the LETKF, an error vector representing the forecast residual is estimated for each ensemble member."

14. Line 12: The term "proxy" could mean a lot of different things. Clarify the nature of updates, whether that's real data assimilation experiment or an OSSE.

We have changed this statement to read "A spatial cross-validation strategy is used to assess the ability of the method to spread the correction along the river network to ungauged locations" (lines 12-13).

- 15. Line 160: I would use "cycled" instead of "iterated" Thank you. This has been changed.
- 16. Line 160: Replace "at each timestep" with "at each observation time"

  We use "timestep" rather than "observation time" as the analysis times are dictated by the availability of the precomputed hindcast data as well as by the availability of observations. For all with the base and this to "extend the act timester for which above retirements.

availability of the precomputed hindcast data as well as by the availability of observations. For clarity, we have changed this to "at each hindcast timestep for which observations are available" (lines 161-162)

17. Line 178: Replace "weights" with "weighs"

Thank you. We have changed this sentence to (lines 177-179): "The Kalman gain matrix determines the impact of the innovation vector in the update step. The respective uncertainties of the prior modelled state and the observations determine their weight within the LETKF."

18. There are too many "see section xxx". This made navigation frustrating; I kept going back and forth. Consider restructuring for better flow.

We have removed cross-references where we deem them to add complexity rather than improve the clarity of the manuscript e.g., as a signpost for the reader for the more novel components of the method (see comment 8 in RC2).

19. The word "improved" is overused in my opinion. Consider other synonyms "enhanced", "refined", ...

We have reworded sentences where appropriate to be more specific about the effect being described.

20. Explain technical terms more clearly, for instance "spatiotemporal consistency" Please see comment 3.

**References**

Emery, C. M., David, C. H., Andreadis, K. M., Turmon, M. J., Reager, J. T., Hobbs, J. M., ... & Rodell, M. (2020). Underlying fundamentals of Kalman filtering for river network modeling. *Journal of Hydrometeorology*, 21(3), 453-474. https://doi.org/10.1175/JHM-D-19-0084.1

Ide, K., Courtier, P., Ghil, M., & Lorenc, A. C. (1997). Unified notation for data assimilation: Operational, sequential and variational (special issue data assimilation in meteorology and oceanography: Theory and practice). *Journal of the Meteorological Society of Japan. Ser. II*, 75(1B), 181-189. https://doi.org/10.2151/jmsj1965.75.1B\_181

Kotsuki, S., Ota, Y., & Miyoshi, T. (2017). Adaptive covariance relaxation methods for ensemble data assimilation: Experiments in the real atmosphere. *Quarterly Journal of the Royal Meteorological Society*, *143*(705), 2001-2015. <a href="https://doi.org/10.1002/qj.3060">https://doi.org/10.1002/qj.3060</a>

Martin, M. J. (2001). *Data assimilation in ocean circulation models with systematic errors* (Doctoral dissertation, University of Reading).

Pauwels, V. R., Hendricks Franssen, H. J., & De Lannoy, G. J. (2020). Evaluation of State and Bias Estimates for Assimilation of SMOS Retrievals Into Conceptual Rainfall-Runoff Models. *Frontiers in Water*, *2*, 4. <a href="https://doi.org/10.3389/frwa.2020.00004">https://doi.org/10.3389/frwa.2020.00004</a>

Rasmussen, J., Madsen, H., Jensen, K. H., & Refsgaard, J. C. (2016). Data assimilation in integrated hydrological modelling in the presence of observation bias. *Hydrology and earth system sciences*, *20*(5), 2103-2118. https://doi.org/10.5194/hess-20-2103-2016

Reichle, R. H., D. B. McLaughlin, and D. Entekhabi, 2002: Hydrologic Data Assimilation with the Ensemble Kalman Filter. *Mon. Wea. Rev.*, 130, 103–114, <a href="https://doi.org/10.1175/1520-0493(2002)130<0103:HDAWTE>2.0.CO;2.">https://doi.org/10.1175/1520-0493(2002)130<0103:HDAWTE>2.0.CO;2.</a>

Ridler, M. E., Zhang, D., Madsen, H., Kidmose, J., Refsgaard, J. C., & Jensen, K. H. (2018). Bias-aware data assimilation in integrated hydrological modelling. *Hydrology Research*, *49*(4), 989-1004. https://doi.org/10.2166/nh.2017.117

Valler, V., Franke, J., & Brönnimann, S. (2019). Impact of different estimations of the background-error covariance matrix on climate reconstructions based on data assimilation. *Climate of the Past*, *15*(4), 1427-1441. https://doi.org/10.5194/cp-15-1427-2019

Zhang, F., Snyder, C., & Sun, J. (2004). Impacts of initial estimate and observation availability on convective-scale data assimilation with an ensemble Kalman filter. *Monthly Weather Review*, *132*(5), 1238-1253. https://doi.org/10.1175/1520-0493(2004)132<1238:IOIEAO>2.0.CO;2

Response to RC2 for hess-2024-3989: Matthews, G., et al. Error-correction across gauged and ungauged locations: A data assimilation-inspired approach to post-processing river discharge forecasts

We thank the reviewer for their insightful comments and helpful suggestions which we believe will greatly strengthen the evaluation and discussion of the new method. The reviewer's comments have been summarised and numbered for clarity. The authors' responses are in blue. Section and line numbers refer to the revised manuscript.

- The paper is too long, in particular the description of the methods.
   We have shortened the descriptions of methods by restructuring and condensing the
   descriptions in Sections 3.1, 5.2, and 5.3. Addressing some of the comments from both
   reviewers such as comment 1, 3, and 12 from RC1 have also reduced sections of the
   manuscript.
- 2. I am a bit concerned about how applicable these methods are outside the case study attempted (see specific comments below). For example, the use of a very large catchment is likely to allow the authors to make simplifying assumptions such as that residuals will be normally distributed, or that errors can be characterised using a 10-day window. I would be interested in some discussion of how generalisable these methods are.
  Thank you for this comment. We have extended the discussion on the generalisability of the method to include the estimation of the initial error ensemble (where the 10-day window is used; lines 660-670) and the assumption of Gaussianity (648-659). See comments 5, 6, and 18.
- 3. L58 "ensemble Kalman Filters are common data assimilation methods for hydrological applications" this is true for hydrological research, but (to me at least) it remains a curiosity as to why data assimilation within hydrological models including with ensemble Kalman Filters remains to my knowledge quite rare in operational streamflow forecasting systems.
  This is a good point. We have specified in "hydrological research applications" and have added the following to the introduction (lines 62-68) "Whilst many studies have shown the benefits of data assimilation for hydrological forecasting (Tanguy et al., 2025; Valdez et al., 2022; Piazzi et al., 2021), the process is rare in operational systems (Pechlivanidis et al., 2025), particularly in large-scale systems (Wu et al., 2020). This limited uptake is partly due to data latency issues (WMO, 2024), time constraints, and the potential impact on the interpretation of the forecasts (e.g., thresholds based on model climatology may no longer be consistent; Emerton et al., 2016). Additionally, the benefit of data assimilation at longer lead-times is uncertain (e.g., Valdez et al., 2022). In this paper, we leverage key advantages of data assimilation—such as the ability to propagate observational information to ungauged locations—within a post-processing framework that is more readily integrated into operational systems."
- 4. L90 "Hydrological ensemble forecasts consist of N potential realizations referred to as ensemble members" I think it would be good to state explicitly which variable(s) you are discussing here, as it wasn't clear to me I'm assuming streamflow (or runoff, as it's on a grid?)? The variable of interest is streamflow or river discharge. We have changed this to "The hydrological ensemble forecasts consist of N potential realizations of future river discharge, referred to as ensemble members" (lines 96-97).

- 5. L107 I would have thought with a strongly skewed (and potentially zero bounded) variable like streamflow, an additive error only generally holds after a normalising transformation has been applied (and, if applicable, zero values have been dealt with).
  We have added normalising transformations to the discussion (see comments 2 and 18). The reviewer is correct that the assumption of Gaussianity limits the applicability of an additive error by occasionally resulting in negative discharges. We deal with negative discharge values as described in Section 4.1. The impact and potential solutions are discussed in Sections 7.2 and 8.
- 6. L112 Similar to the above criticism at L107, Equation 6 appears to assume that errors are normal and homoscedastic. If my understanding of what is being assimilated is correct, this is highly unlikely to hold for streamflow, for which residuals are almost always non-normal and heteroscedastic. See e.g. Smith et al. 2015, among many others.
  We agree that streamflow residuals are often non-Gaussian and heteroscedastic. Our framework does employ updates based on Gaussian assumptions as we use the LETKF, but the resulting distribution is not necessarily Gaussian (Reichle et al., 2002). However, we do not assume homoscedasticity: the ensemble spread evolves dynamically and reflects state-dependent and lead-time-dependent error variability. We have changed the description of the forecast correction to make this more clear (lines 110-120).
- 7. L145 "we adopt the common assumption that the error is constant" I would not have said this is common. I would say it's much more common to use autoregressive models (often AR1) to describe the autocorrelation between residuals in streamflow. I understand why this is a pragmatic simplification, but errors often do change with lead time as the value of forecast information decays.
  Thank you for the comment. We have added the following to Section 3.1 (lines 146-148): "As the
  - Thank you for the comment. We have added the following to Section 3.1 (lines 146-148): "As the true evolution of the error vectors at all grid-boxes is unknown, we assume a simple persistence model, such that  $\mathbf{b}_k^{(i)} = \mathbf{b}_{k-1}^{(i)}$ . This is a common assumption used in state augmentation (Pauwels et al., 2020; Rasmussen et al., 2016; Ridler et al., 2018; Martin, 2001)." We have also added a discussion of this assumption to Section 8 (lines 671-678).
- 8. L149 "define the propagation" I'm not sure what 'propagation' means here, given the error is assumed constant in time. Can the authors clarify? Nevermind the authors do this in Section 3.2! The authors may want to flag that the explanation for this is coming.

  We have signposted that the propagation equation defined on line 147 is for use in the LETKF described in Section 3.2
- 9. L180 "(see Eqs. (8) and (9) in Bell et al., 2004)" I feel that if the authors need to specify equations from another study to describe these methods, the equations should be present in the paper (in an appendix is fine) especially Eq (9) of Bell et al. which the authors later describe as 'key' to the method. (Unless they are included later?)

  We have added the decomposition of the Kalman gain matrix as an appendix (Appendix A)
- 10. Figure 1 this is a really nice, clarifying figure. Thank you!
- 11. L223 "We enforce non-negativity by further adjusting the error ensemble members after the LETKF update step (Fig 1)." This indicates that zero values are present in output state, indicating

that errors are not continuously distributed. I realise not everyone handles zeros, but it would be good to acknowledge the limitation of this assumption (as noted above).

We have extended the discussion regarding the assumption of Gaussianity for river-discharge. See comment 5.

- 12. L265 "Eq. 4.10 in Gaspari and Cohn (1999)" I think the authors should include this equation, as well as discussing (briefly) why they thought the form of this equation appropriate for this task. The regionalisation of errors is in my view the major contribution of the paper.

  The Gaspari and Cohn correlation function is a commonly used localisation function in data assimilation as it smoothly decreases to a definable radius. We have added the Gaspari and Cohn function as an appendix (Appendix B).
- 13. L272 "We propose instead for the localisation length scale to be defined as the maximum distance between any grid point and its closest observation." This seems like a sensible choice. Thank you.
- 14. L325 "(here 10 days)" This is a long period over which to assess an error some use periods of this length for bias correction (e.g. Bennett et al. 2021). I'm assuming this really only works for larger catchments where rivers have slower varying errors; I would have thought for small headwater gauges shorter periods would be more appropriate. It also explains why errors are assumed not to vary with lead time, above. This is all fine, but the authors may wish to mention this in their discussion.

This is a good point. The 10-day period is used to generate the initial error ensemble mean. This initial estimate is not used to correct the river discharge ensembles directly, but rather to provide a starting point for the LETKF. The LETKF then updates the error ensemble at each timestep. We have updated Figure 1 to better clarify this process.

We selected a 10-day period to capture the consistent biases of the hydrological model but also to allow for seasonal/dynamic variation in this bias. We agree that shorter periods may be more appropriate for smaller, fast-responding catchments and have added this to the discussion (lines 660-670).

15.

- a. L408 "we assume that the observation errors from different gauge stations are uncorrelated" I'm not suggesting a change here, and I think this is a reasonable suggestion without additional information. But I suspect the long-range nature of the errors (a 10-day period) may undermine the assumption somewhat.
  The observation errors arise due to instrument uncertainty, observation processing, observation operator error and scale mismatch between the observations and the model resolution. Observation errors are also assumed to be uncorrelated with the prior errors which is a standard assumption in data assimilation. We have clarified this in lines 379-381 and lines 383-384.
- b. I'm also curious what happens when errors are propagated in space: what happens when you get a point equidistant (or close to equidistant) from two gauges, and the errors from the two gauges interact in some way (e.g. cancel each other, or sum). The Kalman gain matrix governs the spreading of observation information in space. A weighted mean is calculated. The weight of an observation is determined by the crosscovariances between the hindcast and error ensembles, the distance from the observation (via the localisation), and the uncertainty in the observation itself. The left and central panels in Figure 2 show single observation experiments, indicating how information from one observation is spread spatially. The panels on the right of Figure 2 show how observation information is propagated when all available observations are assimilated. In Section 8, we have added a discussion on the potential benefits of future work using a block cross-validation to explore the impact of observation density and location (lines 679-683).

16. L438 "forecast mean is decreased by the proposed method we use the Normalised Mean Absolute Error" It's preferable to apply measures of absolute error to the ensemble median. See, e.g., Taggart (2022).

Thank you for this very helpful comment. We have changed Figure 6, and corresponding discussion (lines 584-588) to use the RMSE.

- 17. L526 "However, this assumption is necessary to propagate the hindcast to the next time step without the use of a hydrological model (Section 3.1)." Perhaps, but one application of ensemble predictions is to sum ensemble members through time (e.g. to assess cumulative inflows to reservoirs). From this figure, it seems this would result in highly unreliable accumulations. This may not be an application of EFAS (I don't know), but if the method is to have more general applicability this is a serious weakness.
  - The hydrographs shown in Figures 4b and 4e are not the final hydrographs resulting from this method but instead are intermediate steps used to investigate the impact of the methodological assumptions made. We have restructured Section 7 to make a clearer division between the results that show how the method works and the results that show the skill of the resulting ensemble (see comment 3 from RC1).
- 18. L660 "Future work could look into applying anamorphosis to make the ensemble distribution more Gaussian-like" I'm not familiar with the concept of anamorphosis, but a conventional way of doing this is to use normalising transformations, of which many are available for hydrological variables.

Anamorphosis is very similar to normalising transforms used in hydrology. We have added the use of normalising transformations to this paragraph.

19-23. Typos and grammatical errors.

These typos have all been corrected. Many thanks to the reviewer for catching them!

**References**

Martin, M. J. (2001). *Data assimilation in ocean circulation models with systematic errors* (Doctoral dissertation, University of Reading).

Pauwels, V. R., Hendricks Franssen, H. J., & De Lannoy, G. J. (2020). Evaluation of State and Bias Estimates for Assimilation of SMOS Retrievals Into Conceptual Rainfall-Runoff Models. *Frontiers in Water*, *2*, 4. https://doi.org/10.3389/frwa.2020.00004

Rasmussen, J., Madsen, H., Jensen, K. H., & Refsgaard, J. C. (2016). Data assimilation in integrated hydrological modelling in the presence of observation bias. *Hydrology and earth system sciences*, *20*(5), 2103-2118. <a href="https://doi.org/10.5194/hess-20-2103-2016">https://doi.org/10.5194/hess-20-2103-2016</a>

Reichle, R. H., D. B. McLaughlin, and D. Entekhabi, 2002: Hydrologic Data Assimilation with the Ensemble Kalman Filter. *Mon. Wea. Rev.*, **130**, 103–114, <a href="https://doi.org/10.1175/1520-0493(2002)130<0103:HDAWTE>2.0.CO;2">https://doi.org/10.1175/1520-0493(2002)130<0103:HDAWTE>2.0.CO;2</a>.

Ridler, M. E., Zhang, D., Madsen, H., Kidmose, J., Refsgaard, J. C., & Jensen, K. H. (2018). Bias-aware data assimilation in integrated hydrological modelling. *Hydrology Research*, *49*(4), 989-1004. https://doi.org/10.2166/nh.2017.117

---

## Author Response (AR2)

Response to reviewer's report for hess-2024-3989: Matthews, G., et al. Error-correction across gauged and ungauged locations: A data assimilation-inspired approach to post-processing river discharge forecasts

We thank the reviewer for providing a second review of this manuscript and once again providing thoughtful comments and suggestions to improve the clarity of the manuscript. The authors' responses are in blue. Line number, sections, and figures refer to revised manuscript.

In addition to the reviewers' suggestions, we have corrected typos identified in our own review of the manuscript.

**Specific comments**

1. L84 "Therefore, we refer to these ensembles as hindcasts for clarity." Sorry - I meant to pick this up in the previous revision: what is meant by 'these' here? It appears that here 'hindcasts' refers to something other than 'ensembles of river discharge that we are error correcting', as defined in the opening sentence of this paragraph. Please clarify.

The "hindcasts" do refer to the ensembles of river discharge. In this paper, the ensembles are operational EFAS ensemble forecasts from 2021. However, when we apply the error-correction method we use observations from the forecast period. This is not possible in an operational system as these observations would be in the future. We therefore chose to use the term hindcast to highlight that these ensembles are not valid forecasts. We have added this clarification to lines 88-91:

"In this paper, these ensembles are past operational EFAS forecasts (see Section 6.1). However, when we perform the error-correction we use observations that are available within the forecast (hindcast) period. Observations are not available during the forecast period in an operational system, since these timesteps are in the future. Therefore, we refer to these river discharge ensembles as hindcasts to indicate that the ensembles are not valid forecasts."

2. L199 Figure 1: In the plot accompanying Step 4,  $b(i)_{(k-1)}$  appears to be smaller than  $b(i)_{k}$ . I'm not sure if this is an error, but on face of it this seems to contradict L147 "we assume a simple persistence model, such that  $b(i)_{k} = b(i)_{(k-1)}$ ". Should the figure be showing "b^hat^{(i)a}\_k"?

Thank you for spotting this error. Yes, the figure should show "b^hat^{(i)a}\_k". Figure 1 has been updated.

3. L220 Para starting with "The Kalman filter is not constrained to enforce non-negativity..." thanks for clarifying this. I would imagine that if corrections are applied to the falling limb of the hydrograph (i.e. where the error is computed at high flow and

propagated to low flow) flows would be corrected to zero quite often, especially as  $b(i)_k = b(i)_{(k-1)}$ . This problem would get worse if the method is used to correct multiple lead times. I'm not suggesting a change here - the authors have stated what they have done clearly (at least, if I've understood this correctly!) - but I'd note this is likely to be a serious short-coming for operational deployment, particularly in flashier catchments where flows can vary rapidly over short periods of time.

The error vector is updated at each lead-time with the assumption of a constant error (i.e.,  $b(i)_k = b(i)_{(k-1)}$ ) only assumed during the propagation step of the LETKF. However, the reviewer is correct that the method does struggle to update the error vector correctly for the falling limb of hydrographs as shown in Fig. 5b. This is due to the small ensemble spread (as discussed in lines 584-586). Potential solutions to address the negative discharge issue are discussed in Section 8. The need to correct negative discharge is most often due to the spread of the ensemble not being corrected sufficiently (see lines 622-627) and to the non-Gaussian distribution of river discharge (see lines 658-663).

4. L234 "b^hat^{i}\_k" I think this should be "b^hat^{(i)a}\_k"?

Thank you for spotting this typo, it has been fixed.

5. L458 A useful addition to this plot would be the location of the two gauges being discussed, plotted on the right-hand map.

Thank you for this suggestion. The map has been updated.

- 6. L520 i) "This assumption does ensure the analysis hindcast component is always physically plausible" Not necessarily suggesting a change here, but I could not follow why this this enforces non-negativity. ii) The other question I have here what is causing the update is changing with lead time? Is it that the covariance matrix changes because of changes in the raw ensemble? Or have you assumed the availability of observations (from the statement at L89 in the introduction: "when we perform the error-correction we use observations that are available within the forecast (hindcast) period")?
  - i) The assumption made is that the precomputed hindcast ensemble is a good approximation for the analysis hindcast component. The precomputed hindcast ensemble is the raw output of the LISFLOOD hydrological model and is constrained by the model's physics. I have made this link to non-negative river discharge more explicit (lines 521-522).: "e.g., the river discharge is always positive as this is a constraint within LISFLOOD".
  - ii) The reviewer is correct that we have assumed the availability of observations, and hence we therefore refer to the river discharge ensemble as hindcasts as they are not valid forecasts after the error-correction (see comment 1). As the reviewer notes the updates at each lead-time are also impacted by the

changing covariance matrix of the raw ensemble. The impact of the lead-time dependent updates of the error vectors is discussed in Section 7.1.2.

7. L538 "As demonstrated in Figs. 4c and 4f, this can result in the error ensemble spread being large for the rising limb of an event and smaller for the falling limb." I would say this is a desirable property for a skewed variable, so long as the ensemble spread is still reliable.

We agree with the reviewer that if the ensemble spread is reliable then this property of the ensembles is beneficial. However, the ensemble mean can still be biased. The update of the error vectors is impacted by the spread of the ensemble, resulting in very small updates to the error vector when the spread is narrow (as shown in Fig. 5b). Discussion of the limitation of the method is in lines 646-657.

8. L556 Figure 6. A nice addition to this very informative figure would be the proportion of gauges where corrected forecasts outperformed the raw ensembles for (a)-(d), now that the histograms have been removed. It's sometime hard to make out how many stations have black rings around them or not - e.g. in (b) - so this would neatly summarise this information.

Thank you for this suggestion. The proportion of degraded gauges has been added to the figure caption.

9. L586 "The decrease in N-RMSE, despite an increase in mean bias, suggests that the error-corrected ensembles consistently underestimate flow, while the raw hindcast ensemble fluctuates more between under- and overestimation, which can compensate for each other in the mean bias metric." I'm not sure I agree with this interpretation. 6(b) shows that while there is slight tendency to underestimate flows at some gauges, many sites are unbiased and some have positive biases. Measures of mean squared error applied to skewed variables like streamflow tend to emphasise errors at high flow; better performance at high flow does not suggest a more general tendency to underestimate flow.

Thank you for highlighting this error. The reviewer is correct. This paragraph should have been updated when the NMAE was replaced with the N-RMSE. We have revised the text (lines 587-598):

"Overall, the error-corrected ensemble reduces the N-RMSE but there are 14 stations where the skill is reduced. Typically, these stations are on the upstream reaches of their respective rivers (Fig. 6d; see discussion on correlation). Interestingly, the N-RMSE does not follow the same spatial pattern as the mean bias. This divergence indicates that the correction method is more effective at reducing large errors than at addressing systematic biases. One possible explanation is that the error vectors adjust too slowly to changes in forecast errors between time steps. This slow adjustment is particularly problematic when errors fluctuate around 0 m3s-1, since alternating positive and negative

deviations may not be corrected quickly enough and can accumulate into a worsening mean bias. When the error magnitude is large, the gradual adjustment is less detrimental because the sign of the error is usually captured correctly even if its magnitude is not. However, at upstream stations, where rivers are smaller and respond 595 more quickly to rainfall, large errors often persist for shorter durations, making the slow adjustment of the error vectors more detrimental. This likely contributes to the increase in N-RMSE observed at these upstream stations. Further development of the method—for example, allowing the error vectors to evolve during the propagation step of the LETKF in addition to the update step—could enable faster adaptation to changing forecast errors."

10. L596 "7.2.2 Skill of the ensemble distribution" and Fig 7. It wasn't clear to me which set of predictions is used to generate the plots discussed in this section. Is it the leave-one-gauge out cross-validated predictions (which would be preferable, as the main contribution of the paper is for error predictions in ungauged regions)? Please specify.

Yes, it is the leave-one-gauge out cross-validated predictions that are used. This is stated in lines 545-546.

11. L658 "A transformation between river discharge and specific discharge (river discharge divided by upstream area) could be used to ensure that the ensemble covariances more accurately represent the true relationship between locations." Ok, and I'm not suggesting a change here, but this could mean that more weight is given to upstream gauges in the error analyses, as errors in the timing and location of rainfall tend to cancel over larger areas, resulting in relatively larger errors in headwaters. Using specific discharge would reduce the dominance of larger rivers in the error update, since their larger ensemble variances would have less influence, and the update would instead rely more on the correlation between gauged and ungauged locations. However, as the reviewer notes, a consequence of using the specific discharge could be that upstream gauges are given disproportionately high weights. A study of the benefits and drawbacks of specific discharge, as well as other possible transformations, is left for future work as discussed in lines 661-669.

**Typos etc.**

- 12. Figure 1: "Step 3: Update the error ensemble at by assimilating observations" either there is something missing after 'at' (perhaps 'k+1'?) or delete 'at'. Done. Thank you.
- 13. L261 "Appendix ??" missing cross-reference

The cross-reference has been fixed.

14. L368 "The minimum value across the stations is  $0.516 \, \text{m3s-1}$  and the maximum value is  $7662.917 \, \text{m3s-1}$ ." suggest rounding these numbers: "The minimum value across the stations is <1 m3s-1 and the maximum value is  $7663 \, \text{m3s-1}$ ."

Done. Thank you.

15. L489 "with which the correlation" should be "for which the correlation"

Corrected. Thank you.

16. L491 "the correlations begins" should be "the correlations begin"

Corrected. Thank you.

17. L519 "although, the perturbations" delete comma

Removed.